# Neural dynamics of semantic categorization in semantic variant of primary progressive aphasia

V Borghesani[1]*, CL Dale[2], S Lukic[1], LBN Hinkley[2], M Lauricella[1], W Shwe[1], D Mizuiri[2], S Honma[2], Z Miller[1], B Miller[1], JF Houde[3], ML Gorno-Tempini[1,4], SS Nagarajan[2,3]

[1]Memory and Aging Center, Department of Neurology, University of California, San Francisco, San Francisco, United States; [2]Department of Radiology and Biomedical Imaging, University of California, San Francisco, San Francisco, United States; [3]Department of Otolaryngology, University of California, San Francisco, San Francisco, United States; [4]Department of Neurology, Dyslexia Center University of California, San Francisco, San Francisco, United States

**Abstract** Semantic representations are processed along a posterior-to-anterior gradient reflecting a shift from perceptual (e.g., *it has eight legs*) to conceptual (e.g., *venomous spiders are rare*) information. One critical region is the anterior temporal lobe (ATL): patients with semantic variant primary progressive aphasia (svPPA), a clinical syndrome associated with ATL neurodegeneration, manifest a deep loss of semantic knowledge. We test the hypothesis that svPPA patients perform semantic tasks by over-recruiting areas implicated in perceptual processing. We compared MEG recordings of svPPA patients and healthy controls during a categorization task. While behavioral performance did not differ, svPPA patients showed indications of greater activation over bilateral occipital cortices and superior temporal gyrus, and inconsistent engagement of frontal regions. These findings suggest a pervasive reorganization of brain networks in response to ATL neurodegeneration: the loss of this critical hub leads to a dysregulated (semantic) control system, and defective semantic representations are seemingly compensated via enhanced perceptual processing.

*For correspondence:
valentina.borghesani@ucsf.edu

Competing interests: The authors declare that no competing interests exist.

## Introduction

Approaching a greenish, twisted object during a countryside walk, you might have two very different reactions: running away or simply stepping over it. Such a seemingly easy process, that is, telling a *snake* from a *rope*, requires the interplay of multiple cognitive processes relying on different neural substrates. First, the visual input must be analyzed, collecting information on all possibly relevant motor-perceptual features (e.g., *color, sound, movement*). Then, the extracted features must be merged into a unitary concept to allow proper identification (e.g., *it's a rope*). Finally, one can select and perform an appropriate response (e.g., *I'll walk by it*). All the neural computations supporting these processes occur within a few seconds. While the earliest perceptual processing takes place in the occipital cortex, the final stages (i.e., motor programming and execution) entail activation of frontal-parietal structures. The critical intermediate steps, involving the transformation from a visual input to a concept (and its semantic categorization as living vs. nonliving, dangerous vs. harmless), have been linked to the coordinated activity of multiple neural areas (*Clarke and Tyler, 2015*). Functional neuroimaging and neuropsychological research indicate that semantic knowledge is encoded within distributed networks (*Huth et al., 2012*; *Fernandino et al., 2016*), with a few key cortical regions acting as critical hubs (*Ralph et al., 2017*). However, many open questions remain as to the

nature of neural representations and computations in these different areas, and how they dynamically interact.

Prior functional neuroimaging studies suggested that populations of neurons along the ventral occipito-temporal cortex (vOT) tune to ecologically relevant categories leading to a nested representational hierarchy of visual information (*Grill-Spector and Weiner, 2014*), where specialized cortical regions respond preferentially to faces (*Gauthier et al., 2000*; *Kanwisher et al., 1997*), places (*Epstein and Kanwisher, 1998*), bodies and body parts (*Downing et al., 2007*; *Downing and Kanwisher, 2001*), or objects (*Lerner et al., 2001*). Living stimuli appear to recruit lateral portions of vOT, while nonliving stimuli are highlighted in medial regions (*Martin and Chao, 2001*). Multiple organizing principles appear to be responsible for the representational organization of these areas, including agency and visual categorizability (*Thorat et al., 2019*). Overall, semantic representations appear to be processed in a graded fashion along a posterior-to-anterior axis: from perceptual (e.g., *snakes are elongated and legless*) to conceptual information (e.g., *a snake is a carnivorous reptile*) (*Borghesani et al., 2016*; *Peelen and Caramazza, 2012*). Notwithstanding this overall distributed view, different areas have been linked with specific computational roles: from modality-specific nodes in secondary motor and sensory areas to multimodal convergence hubs in associative cortices (*Binder and Desai, 2011*).

Neuropsychological findings corroborate the idea of a distributed yet specialized organization of semantic processing in the brain, supported by the interaction of a perceptual representational system arising along the occipito-temporal pathway, a semantic representational system confined to the anterior temporal lobe (ATL), and a semantic control system supported by fronto-parietal cortices (*Ralph et al., 2017*). For instance, focal lesions in the occipito-temporal pathway are associated with selective impairment for living items and spared performance on nonliving ones (*Blundo et al., 2006*; *Caramazza and Shelton, 1998*; *Laiacona et al., 2003*; *Pietrini et al., 1988*; *Sartori et al., 1993*; *Warrington and Shallice, 1984*) as well as the opposite pattern (*Laiacona and Capitani, 2001*; *Sacchett and Humphreys, 1992*). Moreover, acute brain damage to prefrontal or temporo-parietal cortices in the semantic control system has been linked with semantic aphasia, a clinical syndrome characterized by deficits in tasks requiring manipulations of semantic knowledge (*Jefferies and Lambon Ralph, 2006*).

A powerful clinical model to study the organization of the semantic system is offered by the semantic variant primary progressive aphasia (svPPA or semantic dementia, *Hodges et al., 1992*; *Gorno-Tempini et al., 2004*). This rare syndrome is associated with ATL neurodegeneration as confirmed by the observation of gray matter atrophy (*Collins et al., 2016*), white matter alterations (*Galantucci et al., 2011*), and hypometabolism (*Diehl et al., 2004*), as well as neuropathological findings (*Hodges and Patterson, 2007*). Patients with svPPA present with an array of impairments (e.g., single-word comprehension deficits, surface dyslexia, impaired object knowledge) that can be traced back to a generalized loss of semantic knowledge, often affecting all stimuli modalities and all semantic categories (*Hodges and Patterson, 2007*). Conversely, executive functions and perceptual abilities are relatively preserved. Hence, these patients provide crucial neuropsychological evidence of the role played by the ATL in the storage of semantic representations, and can be leveraged to investigate the breakdown of the semantic system and the resulting compensatory mechanisms.

Pivotal steps forward in understanding the neurocognitive systems underlying semantic (as well as any other human) behaviors are enabled by the iterative, systematic combination of behavioral and neuroimaging data from both healthy controls (HC) and neurological patients (*Price and Friston, 2002*). However, task-based imaging in patients is hampered by specific difficulties (e.g., patients' compliance) and limitations (e.g., performance is not matched and error signals can act as confounds) (*Price et al., 2006*; *Wilson et al., 2018*). To date, very few studies have attempted to deploy functional imaging in rare clinical syndromes such as svPPA, thus it is still not fully clear how structural damage and functional alterations relate to the observed cognitive and behavioral profile. Previous findings suggest that residual semantic abilities come from the recruitment of homologous and perilesional temporal regions, as well as increased functional demands on the semantic control system, that is, parietal/frontal regions (*Maguire et al., 2010*; *Mummery et al., 1999*; *Pineault et al., 2019*; *Viard et al., 2013*; *Wilson et al., 2009*). Recently, magnetoencephalographic (MEG) imaging has proven useful in detecting syndrome-specific network-level abnormalities (*Ranasinghe et al., 2017*; *Sami et al., 2018*) as well as task-related

functional alterations (*Kielar et al., 2018*) in neurodegenerative patients. Critically, it has been suggested that imperfect behavioral compensation can be achieved via reorganization of the dynamic activity in the brain (*Borghesani et al., 2020*): owing to their damage to the ventral, lexico-semantic reading route, svPPA patients appear to over-recruit the dorsal, sublexical/phonological pathway to read not only pseudowords, but also irregular ones.

Here, we test the hypothesis that svPPA patients, burdened with ATL damage, thus lacking access to specific conceptual representations, overemphasize perceptual information as well as over-tax the semantic control system to maintain accurate performance on a semantic categorization task (living vs. nonliving, see *Figure 1a*). Given the shallow semantic nature of the task, we expect comparable performance in patients with svPPA and a group of HC, with the critical differences emerging in neural signatures. Specifically, we expected patients to over-recruit occipital areas, supporting their greater reliance on visual processing.

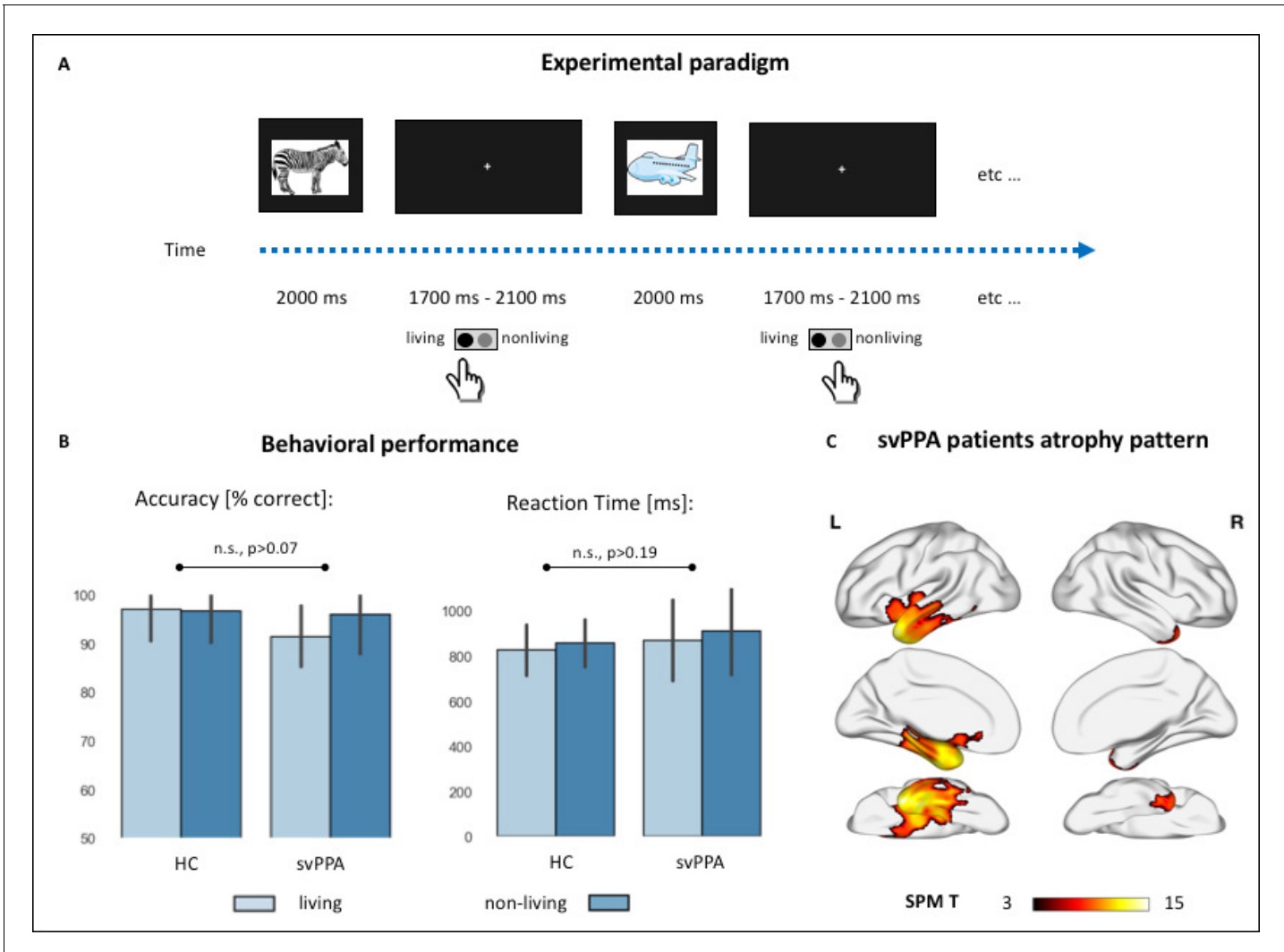

**Figure 1.** Experimental paradigm, behavioral performance, and cortical atrophy. (**A**) Cartoon representation of the experimental setting. Colored drawings were presented for 2 s, with an inter-stimuli interval jittered between 1.7 and 2.1 s. Subjects responded with a button press with their dominant hand. (**B**) Behavioral performance during the semantic categorization tasks in controls and semantic variant primary progressive aphasia (svPPA) patients, across the two stimuli conditions (living vs. nonliving items). There were no statistically significant effects of diagnosis, category, nor their interaction neither in percentage accuracy (healthy controls [HC]: living: 97.1 ± 6.6, nonliving: 96.8 ± 6.6; svPPA: living: 91.5 ± 6.2, nonliving: 95.9 ± 8.1) nor in reaction times (HC: living: 826.3 ± 112.5, nonliving: 856.9 ± 104.4; svPPA: living: 869.8 ± 179.8, nonliving: 911.1 ± 194.45). (**C**) Voxel-based morphometry (VBM)-derived atrophy pattern showing significantly reduced gray matter volumes in svPPA patients' anterior temporal lobes, views from top to bottom: lateral, medial, ventral (thresholded at p<0.05 with family-wise error [FWE] correction, cluster threshold of 100 voxels).

## Results

### Behavioral data and cortical atrophy

Behavioral performance during the MEG scan neither differed between the two cohorts nor between the two stimulus categories. Statistically significant differences were not observed in reaction times (HC: living: 826.3 ± 112.5, nonliving: 856.9 ± 104.4; svPPA: living: 869.8 ± 179.8, nonliving: 911.1 ± 194.45), or accuracy (HC: living: 97.1 ± 6.6, nonliving: 96.8 ± 6.6; svPPA: living: 91.5 ± 6.2, nonliving: 95.9 ± 8.1). Overall, these results indicate that svPPA patients can perform the task as proficiently as healthy elders, an expected finding due to the relatively shallow semantic processing requirements and simple stimuli used in the task (see *Figure 1b*).

Distribution of cortical atrophy in the svPPA cohort is shown in *Figure 1c*. Patients present atrophy in the ATL, involving the temporal pole, the inferior and middle temporal gyrus. This pattern of neurodegeneration, bilateral yet strongly left-lateralized, is consistent with their clinical diagnosis and overall neuropsychological profile (see *Table 1*).

### Time course of neural activity during visual semantic categorization

Within-group analyses of brain activity during the semantic categorization task, relative to pre-stimulus baseline activity levels, are presented for both controls and svPPA patients in *Figure 2*. In brief, following presentation of the images both cohorts showed posterior-to-anterior progression of functional activation across all five frequency bands. In the high-gamma band (63–117 Hz, see *Figure 2a*), we observed bilateral increases in synchronous power starting in the occipital cortex and progressively extending to temporal, parietal, and frontal regions. In the low-gamma band (30–55 Hz, see *Figure 2b*), subjects show heightened synchronization over bilateral occipital cortices, evident early in the svPPA group and only later in HC. Moreover, both groups showed reductions in activity over frontal cortices starting mid-trial. A similar progression of beta (12–30 Hz, see *Figure 2c*) and alpha (8–12 Hz, see *Figure 2d*) band activity revealed significant reductions in synchronous activity for both groups, extending from bilateral occipital cortices to temporal and parietal lobes, and involving progressively larger areas in precentral and superior frontal gyrus. A focus of increased alpha synchrony in anterior cingulate regions, mid-trial, is evident in both groups. Finally, induced theta band (3–7 Hz, see *Figure 2e*) activity revealed progressive increases in synchronous activity over bilateral occipital cortices, a similarly progressive pattern of increased synchronization within frontal regions at an onset window after that of occipital regions, and progressively reduced theta activity relative to baseline levels over parietal and temporal lobes.

Taken together, these stimulus-locked task-induced changes indicate, in both cohorts and across all frequency bands, the expected pattern of visual processing followed by motor response preparation. Notwithstanding the overall similarity in spatiotemporal dynamics, specific activation differences were detected between svPPA patients and HC and are reported below.

### Neural dynamics of semantic categorization in a faulty semantic system

We investigated when, where, and at which frequency svPPA patients differ from HC during semantic categorization of visual stimuli. While the overall pattern of activation across frequencies and time is similar, crucial differences between the two cohorts emerged in the between-group analyses performed in each frequency band. *Table 2* summarizes the temporal windows, peaks of local maxima, and t-values of all clusters isolated by the direct comparison of the two cohorts. *Figure 3* allows appreciation of the spatiotemporal distribution of these clusters at four exemplar time points.

In the high-gamma band, we detected significantly higher synchronization in svPPA patients, relative to controls, over left superior temporal (at both early and late time points) and right frontal (at late time points) cortices (see *Figure 3a*). In the low-gamma band, we observed an extensive spatiotemporal cluster over bilateral occipital cortices with significantly higher synchronized activity in svPPA patients relative to controls. Similarly, small clusters of gamma activity, relatively more desynchronized in HC than svPPA, resulted in an increased gamma synchrony in medial frontal cortices at ~300 ms for the svPPA group (see *Figure 3b*). Overall, the results at high frequencies (30–117 Hz) suggest thus higher activity in svPPA over bilateral occipital and left superior temporal cortices throughout the trial, and right frontal cortices at late time points.

**Table 1.** Demographics and neuropsychological profiles.

Healthy controls and semantic variant of primary progressive aphasia (svPPA) patients, native English speakers, were matched for age, gender, and education. Scores shown are mean (standard deviation). * Indicates values significantly different from controls (p<0.05). MMSE = Mini-Mental State Exam; CDR = Clinical Dementia Rating; PPVT = Picture Vocabulary Test; WAB = Western Aphasia Battery; VOSP = Visual Object and Space Perception Battery.

| | Controls | svPPA |
|---|---|---|
| Demographic | | |
| N | 18 | 18 |
| Age, mean (SD) | 70.7 ± 6.5 | 67.1 ± 6.2 |
| Education, mean (SD) | 17.5 ± 1.8 | 17.9 ± 3.2 |
| Gender, n female | 12 | 9 |
| Handedness, n right | 15 | 15 |
| MMSE (max. 30) | 29.0 ± 1.6 | 24.5 ± 3.8* |
| CDR score | 0.03 ± 0.1 | 0.7 ± 0.4* |
| CDR box score | 0.3 ± 1.2 | 4.0 ± 2.6* |
| Language production | | |
| Boston (object) naming test (15) | 14.7 ± 0.6 | 5.4 ± 3.7* |
| Phonemic (D-letter) fluency | 15.7 ± 5.8 | 9.1 ± 4.3* |
| Semantic (animal) fluency | 23.4 ± 3.9 | 9.3 ± 4.1* |
| Language comprehension | | |
| PPVT (max. 16) | – | 9.4 ± 3.2 |
| WAB auditory word recognition (60) | – | 56.5 ± 4.2 |
| WAB sequential command (100) | – | 70.7 ± 14.3 |
| Digit span forwards | 7.1 ± 1.1 | 6.4 ± 1.2 |
| Reading | | |
| Arizona reading total (max. 36) | 35.6 ± 0.5 | 30.5 ± 3.7* |
| Regular high-frequency words (9) | 9 ± 0.0 | 8.8 ± 0.4 |
| Regular low-frequency words (9) | 8.9 ± 0.2 | 8.3 ± 1.2 |
| Irregular high-frequency words (9) | 8.9 ± 0.3 | 7.7 ± 0.6 |
| Irregular low-frequency words (9) | 8.8 ± 0.4 | 5.7 ± 2.3 |
| Pseudowords (18) | 15.8 ± 2.7 | 15.2 ± 2.2 |
| Spelling | | |
| Arizona spelling total (max. 20) | 18.1 ± 1.6 | 13.1 ± 4.0* |
| Regular high-frequency words (5) | 5 ± 0.0 | 4.4 ± 0.9 |
| Regular low-frequency words (5) | 4.5 ± 0.6 | 4.1 ± 0.8 |
| Irregular high-frequency words (5) | 4.1 ± 0.9 | 2.1 ± 1.6 |
| Irregular low-frequency words (5) | 4.5 ± 0.5 | 2.6 ± 1.6 |
| Pseudowords (10) | 8.8 ± 1.3 | 8.1 ± 2.6 |
| Famous faces – spontaneous naming (max. 16) | 12.4 ± 3.4 | 2.9 ± 2.4* |
| Famous faces – face recognition (max 20) | 18.4 ± 2.0 | 12.8 ± 6.5* |
| Famous faces short triplets, pictures (max. 10) | 8.9 ± 1.0 | 6.6 ± 2.4 |
| Famous faces short triplets, words (max. 10) | 9.7 ± 0.6 | 7.0 ± 2.0 |
| Working memory/executive functions | | |
| Digit span backwards | 5.4 ± 1.1 | 4.5 ± 1.6* |
| Modified trials (total time) | 25.3 ± 13.6 | 41.9 ± 23.1* |
| Modified trials (# of correct lines) | 13.2 ± 3.2 | 13.2 ± 3.3 |

*Table 1 continued on next page*

*Table 1 continued*

|  | Controls | svPPA |
| --- | --- | --- |
| Design fluency (# of correct designs) | 11.7 ± 3.0 | 7.1 ± 3.4* |
| Visuospatial function |  |  |
| Benson figure copy (17) | 15.7 ± 0.7 | 15.3 ± 1.0 |
| VOSP number location (30) | 9.3 ± 0.9 | 9.0 ± 1.5 |
| Visual memory |  |  |
| Benson figure recall (17) | 12.1 ± 2.4 | 7.1 ± 4.9* |

Between-group contrast in beta band revealed, in svPPA patients, more desynchronization (i.e., more beta suppression) over the left superior temporal gyrus at ~300 ms, while simultaneously displaying less desynchronization in a right middle-frontal cluster (see *Figure 3c*). In the alpha band, svPPA patients showed less desynchronization over left middle temporal gyrus at ~300 ms as well as in later clusters in the right precentral gyrus, left anterior cingulate, and left parahippocampal gyrus (see *Figure 3d*). Finally, in the theta band, significant differences over the left occipital cortex occurred at both early (~100 ms) and late (~500 ms) time points indicating higher synchronization in svPPA patients compared to HC, while the opposite pattern (i.e., higher activity for HC) is observed in a right frontal cluster at ~300 ms (see *Figure 3e*). Overall, the results at low frequencies (3–30 Hz) suggest thus higher activity in svPPA over bilateral occipital and left superior temporal cortices, while indicating less activity in left middle temporal and right frontal regions.

Taken together, these findings suggest that svPPA patients performed the semantic categorization tasks by over-recruiting bilateral occipital cortices and left superior temporal gyrus, while showing less reliance on left middle temporal regions and inconsistent engagement of frontal ones.

## Post hoc region-of-interest analyses

Our first region-of-interest (ROI) post hoc analysis allows visualization, across all frequency bands, of the differences in temporal dynamics between the two cohorts (*Figure 4*). The three a priori defined ROIs cover the theorized perceptual-to-conceptual gradient of information processing along the ventral visual path (*Borghesani and Piazza, 2017*) and include the putative visual spoke (left occipital pole, OCC) and semantic hub (left ATL, *Ralph et al., 2017*). It appears clear that the main difference between svPPA patients and HC is heightened low-gamma activity over the occipital region. Such difference is evident around 100 ms post stimuli onset, peaks around 200 ms, and continues throughout the whole. These findings rule out an explanation of the observed whole brain differences as mere temporal shift or spreading, while highlighting the spatial specificity of the main results.

Our second ROI post hoc analysis allows characterization of the full time-frequency spectrum of both cohorts in two representative voxels (*Figure 5*). Critically, a broad and sustained increase in low-gamma band power is observed in svPPA patients and not in HC, with no traces of cross-spectral leakage between beta and low gamma, or low and high gamma. These findings rule out a possible interpretation of the observed effects in terms of frequency shift or spread, highlighting the spectral specificity of the main results.

## Discussion

This is the first study investigating the spatiotemporal dynamics of semantic categorization of visual stimuli in a cohort of svPPA patients. We provide compelling evidence that, burdened with ATL damage, svPPA patients recruit additional perilesional and distal cortical regions to achieve normal performance on a shallow semantic task. As compared to healthy age-matched controls, svPPA patients showed greater activation over bilateral occipital cortices and superior temporal gyrus, indicating over-reliance on perceptual processing and spared dorsal language networks. Conversely, they showed inconsistent engagement of frontal regions, suggesting less efficient control responses.

These findings have important implications both for current neurocognitive models of the language systems and on the utility of MEG imaging in clinical populations. First, we detect over-recruitment of occipital and superior temporal regions paired with inconsistent engagement of

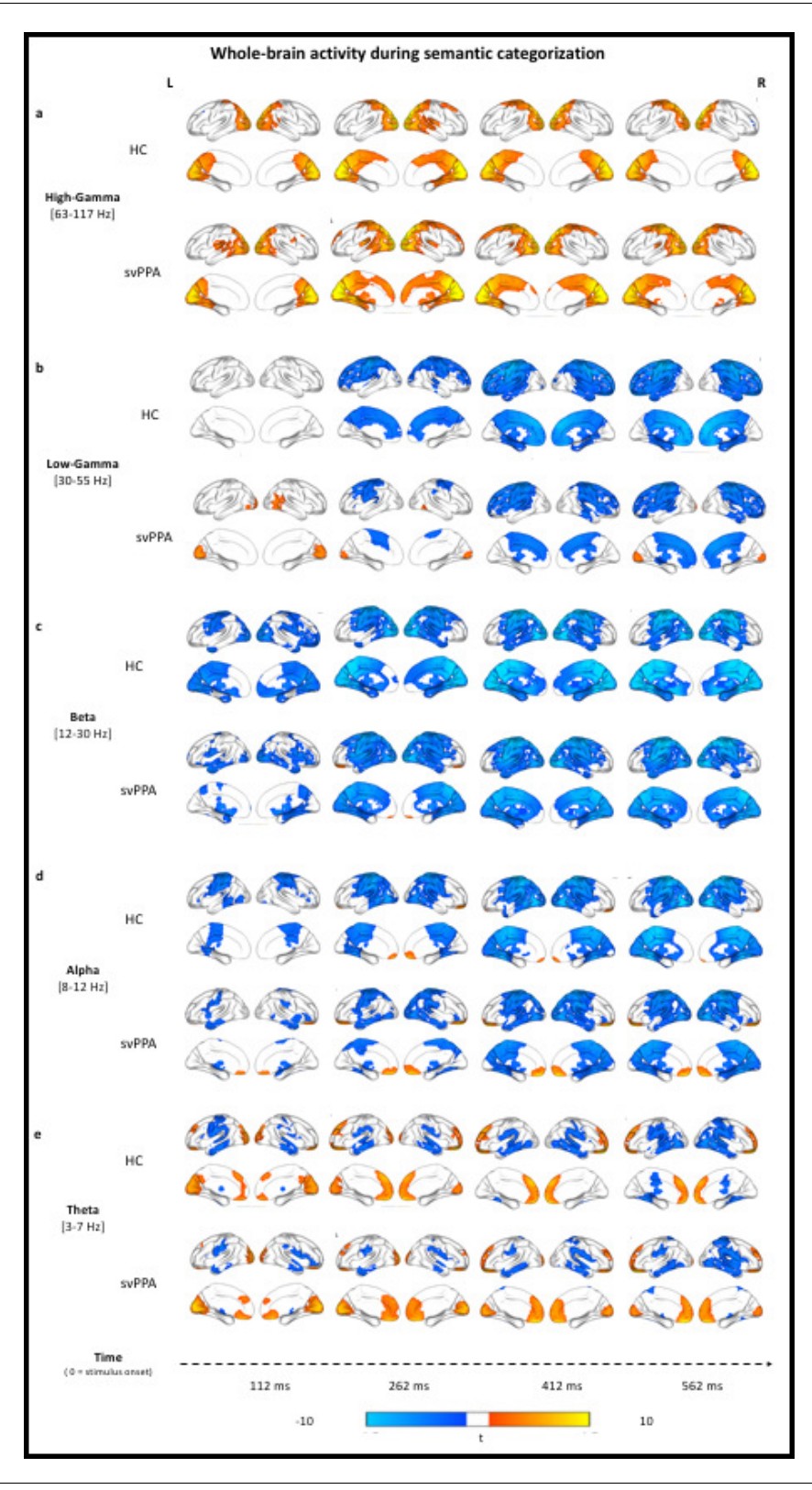

**Figure 2.** Stimulus-locked (0 ms = stimulus onset) within-group analyses of task-related changes in oscillatory power. (**a**) Rendering of the results in the high-gamma band for both controls (healthy controls [HC], upper row) and patients (semantic variant primary progressive aphasia [svPPA], lower row). Cold color = more desynchronization (vs. baseline). Warm color = more synchronization (vs. baseline). (**c-e**) Same as in (**a**) but for the low-gamma, beta,

*Figure 2 continued on next page*

*Figure 2 continued*

alpha, and theta band, respectively. Within-group analyses were performed, with no additional smoothing, on normalized reconstructions using statistical nonparametric mapping (SnPM one-sample, two-tailed t-test against baseline).

frontal areas, where some spatiotemporal clusters suggest heightened activity in patients, others in controls. These results speak to the distributed and dynamic organization of the semantic system, where semantic representations are supported by occipito-temporal cortices and semantic control by fronto-parietal areas. Second, the observation that normal performance can be achieved via altered neural dynamics elucidates the neurocognitive mechanisms that support compensation in neurological patients. Specifically, we contribute to the body of literature illustrating how network-driven neurodegeneration leads to the reorganization of the interplay of various cortical regions.

**Table 2.** Local maxima in Montreal Neurological Institute (MNI) coordinates.

Time window, MNI coordinates, p- and t-value of the local maxima of the different magnetoencephalographic (MEG) whole brain contrasts performed. The spatiotemporal distribution of these clusters at four exemplar time points can be appreciated in *Figure 3*.

| t-test svPPA vs. HC | Time window | Local maxima | | | |
|---|---|---|---|---|---|
| | ms | MNI [x,y,z] | p-value | t-value | |
| Theta band [3–7 Hz] | | | | | |
| Left lingual gyrus | 0–212 | −10.0 −100.0 −10.0 | 0.005 | 3.7 | More ERD in svPPA |
| Left lingual gyrus | 412–612 | −8.5 −100.0 −8.1 | 0.005 | 3.1 | More ERD in svPPA |
| Right medial and superior frontal gyrus | 187–387 | 18.6 61.4 −14.7 | 0.001 | −3.92 | Less ERS in svPPA |
| Alpha band [8–12 Hz] | | | | | |
| Right precentral gyrus | 212–612 | 45.0 −15.0 40.0 | 0.001 | 3.4 | More ERD in svPPA |
| Left middle temporal gyrus | 287–362 | −59.8 −41.6 −1.0 | 0.005 | 2.8 | More ERD in svPPA |
| Bilateral medial and orbital frontal gyrus | 462–612 | −6.2 34.3 −24.9 | 0.001 | 5.1 | More ERS in svPPA |
| Beta band [12–30 Hz] | | | | | |
| Left cingulate cortex | 0–62 | −6.2 −30.3 43.3 | 0.005 | 2.9 | More ERS in svPPA |
| Right medial frontal gyrus | 137–262 | 7.8 56.8 11.7 | 0.001 | 3.6 | More ERS in svPPA |
| Left middle temporal gyrus | 237–362 | −65.0 −20.0 −5.0 | 0.001 | −3.4 | Less ERD in svPPA |
| Left superior frontal gyrus | 587–612 | −21.8 46.7 45.7 | 0.005 | −3.1 | Less ERD in svPPA |
| Low-gamma band [30–55 Hz] | | | | | |
| Left lingual gyrus | 62–612 | −10.1 −98.4 −8.9 | 0.001 | 4.2 | More ERS in svPPA |
| Left inferior occipital gyrus | 362–612 | −34.8 −93.9 2.7 | 0.001 | 4.1 | More ERS in svPPA |
| Right lingual gyrus | 212–437 | 18.2 −89.1 8.3 | 0.005 | 3.4 | More ERS in svPPA |
| Right medial frontal gyrus | 212–412 | 9.3 63.0 2.2 | 0.001 | 3.7 | Less ERD in svPPA |
| Left superior frontal gyrus | 262–462 | −3.8 62.8 14.0 | 0.005 | 3.6 | Less ERD in svPPA |
| High-gamma band [63–117 Hz] | | | | | |
| Left superior frontal gyrus | 62–137 | −36.5 26.6 48.8 | 0.001 | 3.4 | More ERS in svPPA |
| Left superior temporal gyrus | 62–287 | −48.2 −22.3 13.3 | 0.005 | 3 | More ERS in svPPA |
| Left parahippocampal gyrus | 212–312 | −15.5 −27.1 −6.5 | 0.001 | 3.3 | More ERS in svPPA |
| Right medial frontal gyrus | 287–337 | 13.2 70.7 0.6 | 0.005 | 3.2 | More ERS in svPPA |
| Left superior frontal gyrus | 287–612 | −22 68.4 14 | 0.001 | 3.6 | More ERS in svPPA |
| Right superior frontal gyrus | 462–612 | 43.9 54.7 17.2 | 0.001 | 3.9 | More ERS in svPPA |

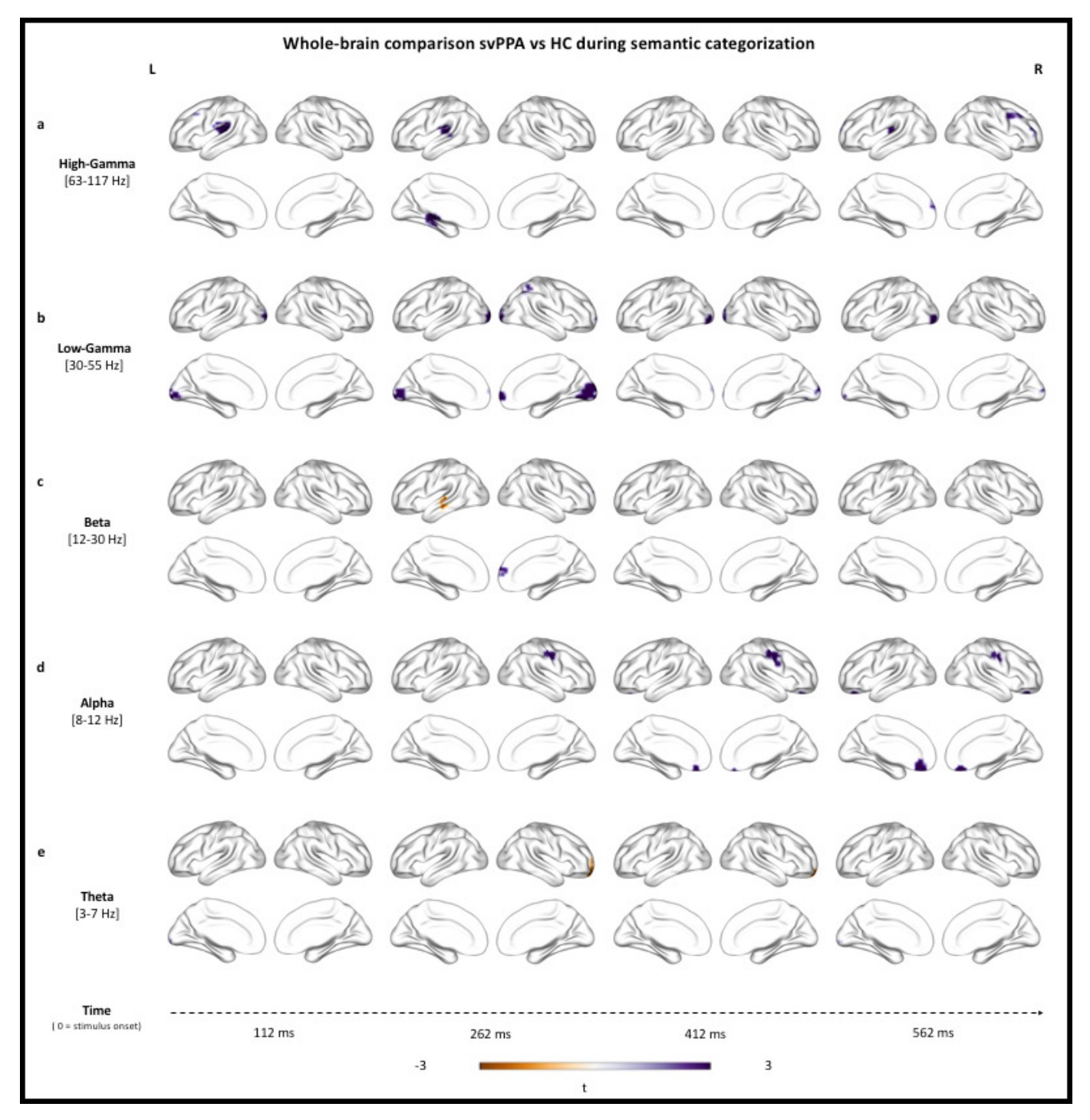

**Figure 3.** Stimulus-locked (0 ms = stimulus onset) between-group analyses of changes in oscillatory power. Rendering of the results in the high-gamma (a), low-gamma (b), beta (c), alpha (d), and theta (e) bands. Purple color = more synchronization in semantic variant primary progressive aphasia (svPPA) (vs. healthy controls [HC]). Brown color = less synchronization in svPPA (vs. HC). *Table 2* summarizes the temporal windows, peaks of local maxima, and t-values of all clusters isolated by the direct comparison of the two cohorts. Between-group analyses were performed, with no additional smoothing, on normalized reconstructions using statistical nonparametric mapping (SnPM two-sample, two-tailed t-test).

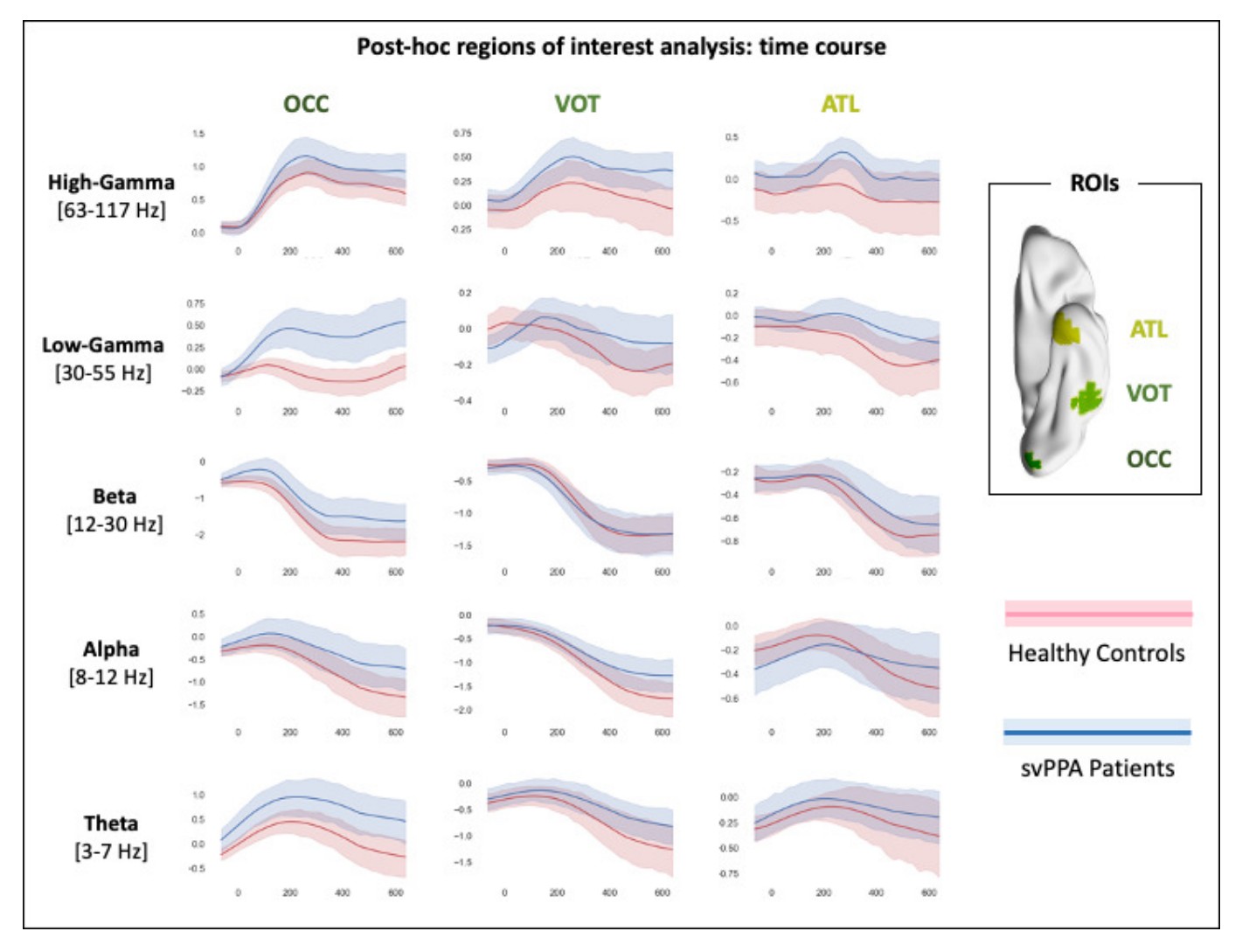

**Figure 4.** Results of the region of interest post hoc analysis. Three regions-of-interest (ROIs) of 20 mm radius were centered on the occipital pole (OCC, Montreal Neurological Institute [MNI]: −10, –94, −16), left ventral occipito-temporal cortex (VOT, MNI: −50, –52, −20), and left ATL (MNI: −30, –6, −40). Pink color represents healthy controls data, light blue svPPA patients. Shaded areas indicate the standard deviation around the group average (i.e., solid line).

## Faulty semantic representations: compensating conceptual loss with perceptual information

Our key finding is that svPPA patients can achieve normal performance in a shallow semantic task by over-relying on perilesional language-related regions (STG), as well as on distal visual (occipital) and executive (frontal) networks. At frequencies spanning low and high gamma bands, svPPA patients show increased activity in occipital and superior temporal cortices relative to their healthy counterparts. Gamma oscillations have been associated with local computations (*Donner and Siegel, 2011*), promoting binding and selective long-range communication (*Hagoort et al., 2004*; *Fries, 2015*), including merging of multimodal semantic information (*van Ackeren et al., 2014*). Results at lower frequencies indicate greater neural activity in svPPA over bilateral occipital and left superior temporal cortices. Theta oscillations have generally been associated with operations over distributed networks, such as those required for lexico-semantic retrieval (*Bastiaansen et al., 2005*; *Bastiaansen et al., 2008*; *Kielar et al., 2015*), integration of unimodal semantic features (*van Ackeren et al., 2014*), and facilitating phase-specific coupling of selective communication

between regions (*Fries, 2015*; *Canolty et al., 2006*). Therefore, in our patients, compensation for faulty semantic representations seems to rely primarily on local and distributed computations in networks associated with perceptual processing.

In principle, the semantic task employed in the current study (i.e., identifying a visually presented object as either a living or nonliving) can be performed by focusing on a few key, distinctive, motor-perceptual features: *if it has eyes and teeth, it is a living being*. Further processing steps, such as would be required for an object identification and naming (i.e., accessing the appropriate lexical label), require the integration of multiple motor perceptual as well as conceptual features (*Borghesani and Piazza, 2017*): *a python is a nonvenomous snake that kills by constriction*. Combining the behavioral data collected during the recordings and outside and the scanner (see Boston Naming Task performance, *Table 1*), it appears clear that HC can recognize (and likely inevitably mentally name) each item, while svPPA patients can only provide the categorical label. Patient data is thus critical in characterizing the division of labor between the distributed set of cortical regions involved in semantic processing. Our findings strongly suggest that ATL damage hampers operation of the semantic representation system, by shattering their conceptual components and thus forcing over-reliance on perceptual features coded in posterior cortices. This is consistent with a growing body of research. For instance, it has been shown that the ability to merge perceptual features into semantic concepts relies on the integrity of the ATL (*Hoffman et al., 2014*), and that ATL damage promotes reliance on perceptual similarities over conceptual ones (*Lambon Ralph et al., 2010*). Moreover, it appears that the more motor-perceptual information is associated with a given concept, the more resilient it is to damage, an advantage that is lost once the disease progresses from ATL to posterior ventral temporal regions (*Hoffman et al., 2012*).

## Faulty semantic representations: overtaxing the semantic control network

Compared to HC, svPPA patients appear to have less activation in the left middle temporal gyrus and to inconsistently engage frontal regions, suggesting that increased demands to the semantic control systems are met by inefficient responses in prefrontal and superior frontal cortices. Comparing the two cohorts across frequency bands, it appears that an enhanced late high-frequency (local neural) response occurs in svPPA, vs. an earlier and lower frequency (long-range connection) response in controls. One speculation for this pattern is that in svPPA, an initial inefficient response in the (semantic) cognitive control network centered on frontal areas leads to a later higher reliance on local activity for (semantic) cognitive control and decision-making processes.

Previous studies demonstrated that object recognition in visual areas is facilitated by prior knowledge (*Bannert and Bartels, 2013*) received via feedback projections from both frontal (*Bar et al., 2006*) and anterior temporal (*Coutanche and Thompson-Schill, 2015*) cortices. Moreover, it has been observed that higher demands for feature integration entail more recurrent activity between fusiform and ATL (*Clarke et al., 2011*). Our study provides a direct contrast between subjects in which both frontal and ATL feedback inputs are preserved (HC), and those in which ATL neurodegeneration forces reliance exclusively on frontal inputs.

Interestingly, the observed temporal dynamics (with the detection of early frontal involvement) are not compatible with a strictly feedforward model of visual stimuli processing. This is in line with recent evidence that recurrent neural models are needed to explain the representational transformations supporting visual information processing (*Gwilliams and King, 2019*; *Kietzmann et al., 2019*).

Thus, taken together, our findings corroborate the idea that the conversion from percept to concept is supported by recurrent loops over fronto-parietal and occipito-temporal regions which have been implicated in, respectively, semantic control and semantic representations (*Chiou et al., 2018*).

## Clinical implications

Our findings corroborate the idea that neurodegeneration leads to the dynamic reorganization of distributed networks (*Agosta et al., 2014*; *Guo et al., 2013*), and that task-based MEG imaging can be instrumental in deepening our understanding of the resulting alterations (*Borghesani et al., 2020*). Ultimately, these efforts will pave the way toward treatment options, as well as better early diagnostic markers as functional changes are known to precede structural ones (*Bonakdarpour et al., 2017*). For instance, our results support previous neuropsychological evidence

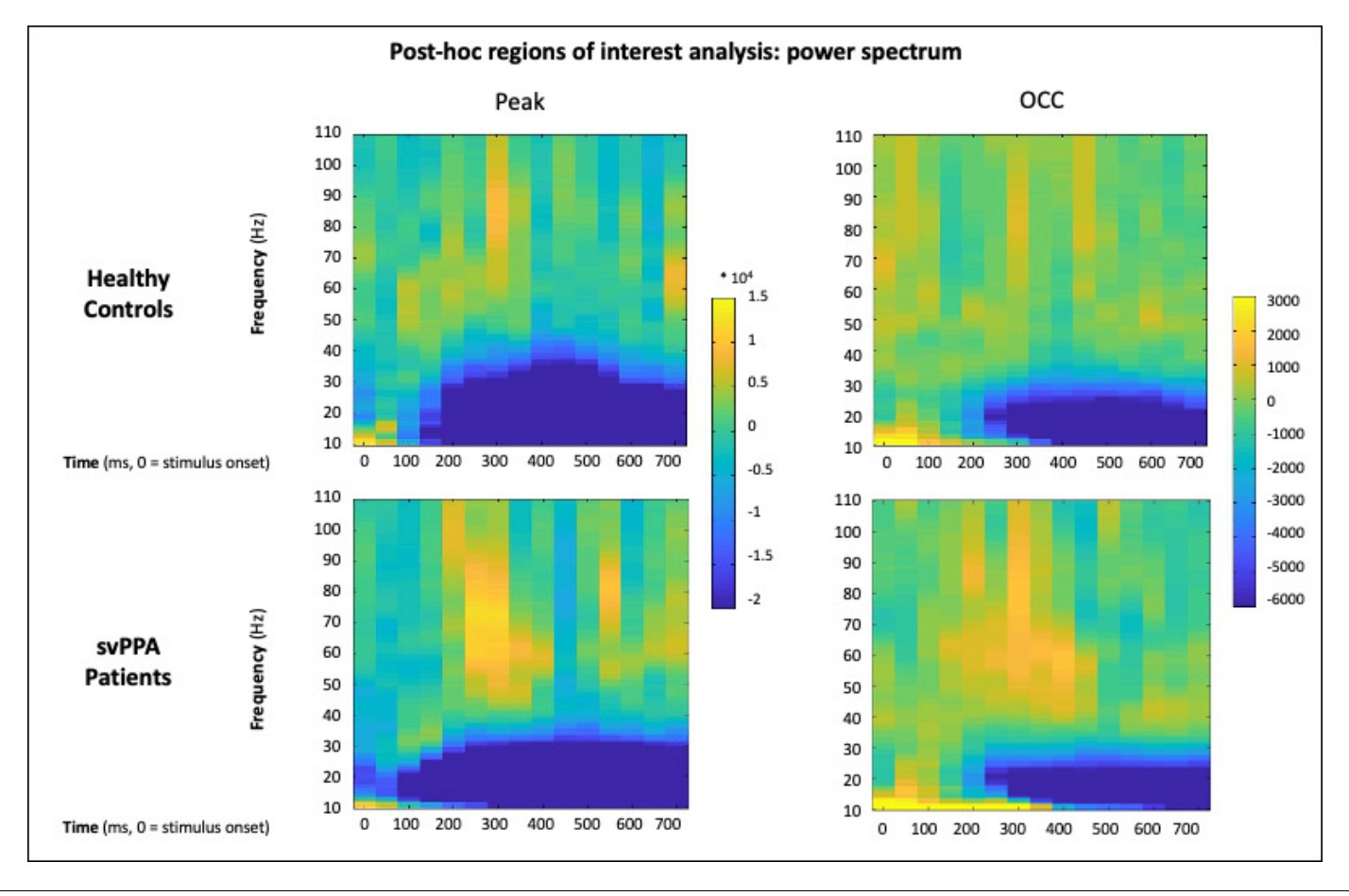

**Figure 5.** Results of the post hoc regions of interest analysis of power changes. Full time-frequency plot of power changes in two representative voxels centered in the peak of activation (as per group results, Montreal Neurological Institute [MNI]: −34.8, –93.9, 2.7) and on the occipital pole (OCC, MNI: −10, –94, −16).

suggesting that the origin of svPPA patients' difficulties during semantic categorization tasks are linked to degraded feature knowledge rather than, as it happens in other FTDs (fronto-temporal dementias), to a deficit of executive processes (*Koenig et al., 2006*).

Our results are in line with prior studies relating svPPA patients' performance on semantic tasks with respect to not only the expected hypoactivation of the left ATL and functionally connected left posterior inferior temporal lobe (*Mummery et al., 1999*), but also based on the patterns of hyperactivations observed in the current study. Heightened activity has been reported in periatrophic left anterior superior temporal gyrus as well as more distant left premotor cortex, and right ATL (*Mummery et al., 1999*; *Pineault et al., 2019*). Individual subject analyses have indicated that patients might attempt different compensatory strategies, which may vary in terms of efficiency and, crucially, would rely on the recruitment of different cortical networks (*Viard et al., 2013*; *Viard et al., 2014*). For instance, studies on reading have associated svPPA patients' imperfect compensation of the semantic deficit (leading to regularization errors) with over-reliance on parietal regions subserving sublexical processes (*Wilson et al., 2009*). Consistently, task-free studies of intrinsic functional networks suggest that the downregulation of damaged neurocognitive systems can be associated with the upregulation of spared ones. In svPPA patients, recent fMRI evidence shows coupling of decreased connectivity in the ventral semantic network with increased connectivity in the dorsal articulatory-phonological one (*Battistella et al., 2019*; *Montembeault et al., 2019*). Additionally, svPPA has been linked with specific spatiotemporal patterns of neuronal synchrony alterations: alpha and beta hypersynchrony in the left posterior superior temporal and adjacent

parietal cortices, and delta-theta hypersynchrony in left posterior temporal/occipital cortices (*Ranasinghe et al., 2017*). Our findings also align with the recent observation that, during reading, svPPA patients can (imperfectly) compensate for their damage to the ventral route by over-recruiting the dorsal one (*Borghesani et al., 2020*). The present findings corroborate thus the idea that neurodegeneration forces the reorganization of the interplay between ventral and dorsal language networks.

Critically, the present functional neuroimaging results and their interpretation rest on the fact that the task allowed engagement of semantic processing in patients in which the semantic system is, by definition, compromised. Contrary to a more challenging task such as naming, patients with svPPA were able to perform the semantic categorization as accurately and fast as HC. Hence, probing the semantic system at the proper level of difficulty (*Wilson et al., 2018*), we avoided the challenging interpretation of activation maps associated with failure to perform a task (*Price et al., 2006*). Our findings thus call for caution when evaluating studies comparing clinical cohorts based solely on behavioral data: failing to detect a difference in performance does not necessarily correspond to similar underlying neurocognitive resources.

## Limitations and future perspectives

The nature of the clinical model we adopted constrains our sample. First, even if ours is the to-date largest cohort of svPPA patients assessed with task-based functional neuroimaging, our sample size is relatively small, owing to the rareness of the disease. We thus have limited statistical power, preventing us from, for instance, further exploring brain-behavior correlations. Second, our subjects (both HC and patients) are older than those reported in previous studies on semantic categorization, cautioning against direct comparisons. While it has been shown that the neural dynamics of visual processing are affected by aging, the reduced and delayed activity observed does not necessarily relate to poorer performance, but rather may be mediated by task difficulty (*Bruffaerts et al., 2019*). Moreover, previous evidence suggests that even if semantic processing remains intact during aging, its neurofunctional organization undergoes changes. For instance, *Lacombe et al., 2015* found that, during a verbal semantic categorization task, older adults exhibited behavioral performance equivalent to that of young adults, but showed less activation of the left inferior parietal cortex and more activation of bilateral temporal cortex. Finally, our task design does not allow further investigation of potential categorical effects. Future studies wishing to investigate representations of living and nonliving items separately will require more trials and stimuli carefully controlled for psycholinguistic variables such as prototypicality and familiarity. Contrary to patients with damage to the vOT due to stroke or herpes simplex encephalitis, svPPA patients usually do not present categorical dissociations (*Moss et al., 2005*). However, deeper investigations of time-resolved neural activity in svPPA could shed light onto the debate on the nature of ATL representations: category-specific deficits might arise from lacunar (rather than generalized) impairment of graded representations (*Lambon Ralph et al., 2007*).

To date, the field lacks proper strategies to deal with tissue undergoing neurodegeneration while attempting to source-localize electrophysiological effects. The first and main issue is that of defining atrophy itself: atrophic tissue will vary subject by subject (and it is not an all-or-none phenomenon) yet a threshold would need to be established. In previous work, we took the parsimonious approach of masking out atrophic regions from group-level statistics to avoid uninterpretable results (e.g., *Borghesani et al., 2020*). In other settings, one option is that of correcting ROI statistics including gray matter volume as covariate (e.g., *Ranasinghe et al., 2017*). Finally, some would simply report whether electrophysiological differences and atrophy maps overlap or not (e.g., *Kielar et al., 2018*). Here, we decided not to mask the ATL and, as a consequence, an apparent signal coming from the atrophic region is observed in svPPA patients (mainly in beta and alpha). Clearly, the underlying issue of how tissue undergoing neurodegeneration affects source modeling is an open problem that requires further exploration.

Finally, our interpretation of higher low gamma as a sign of compensation is speculative and lies on twofold inference: that more low-gamma band in svPPA means more activity, and that more activity means compensation. Previous literature in healthy and pathological aging suggests that higher activation can be associated with compensatory effects or reflect neuropathology (e.g., *Elman et al., 2014*). Critically, when considering progressive phenomena such as neurodegeneration, one has to acknowledge that hyper- and hypoactivations might reflect different stages of

disease and that their relation to behavioral performance might follow a nonlinear U-shaped trajectory (e.g., *Gregory et al., 2017*). Following *Cabeza et al., 2018*, we believe that greater activity can be interpreted as compensation if two criteria are met: (1) there has to be evidence of a 'supply-demand gap' and (2) there has to be evidence of a beneficial effect on cognitive performance. Our data clearly fulfills the first criterion as svPPA patients present insufficient neural resources (i.e., relatively focal ATL atrophy) and suffer from its behavioral consequences (i.e., pervasive semantic loss). Our findings also fulfill the second criterion as svPPA patients are able to perform the task with both accuracy and reaction times comparable with the HC. Thus, we believe that the neural and behavioral evidence we provide is enough to rule out alternatives to compensation such as inefficiency or pathology. While further studies are warranted to shed light onto the relation between neurodegeneration, neurophysiological markers, and behavior, we hold that the most appropriate (albeit speculative) interpretation of our findings is in terms of compensation.

## Conclusions

Combining task-based MEG imaging and a neuropsychological model, we provide novel evidence that faulty semantic representations following ATL damage can be partially circumvented by additional processing in relatively spared occipital and dorsal stream regions. Our results thus inform current neurocognitive models of the semantics system by corroborating the idea that it relies on the dynamic interplay of distributed functional neural networks. Moreover, we highlight how MEG imaging can be leveraged in clinical populations to study compensation mechanisms such as the recruitment of perilesional and distal cortical regions.

# Materials and methods

## Subjects

Eighteen svPPA patients (13 females, 66.9 ± 6.9 years of age) and 18 healthy age-matched controls (11 females, 71.3 ± 6.1 years of age) were recruited through the University of California San Francisco (UCSF) Memory and Aging Center (MAC). All subjects were native speakers and had no contraindications to MEG. Patients met currently published criteria as determined by a team of clinicians based on a detailed medical history, comprehensive neurological and standardized neuropsychological and language evaluations (*Gorno-Tempini et al., 2011*). Besides being diagnosed with svPPA, patients were required to score at least 15 out of 30 on the Mini-Mental Status Exam (MMSE; *Folstein et al., 1975*) and be otherwise sufficiently functional to be scanned. HC were recruited from the UCSF MAC healthy aging cohort, a collection of subjects with normal cognitive and neurological exam and MRI scans without clinically evident strokes. Inclusion criteria required the absence of any psychiatric symptoms or cognitive deficits (i.e., Clinical Dementia Rating – CDR = 0 and MMSE ≥28/30). Demographic information and neuropsychological data are shown in *Table 1*. The study was approved by the UCSF Committee on Human Research and all subjects provided written informed consent (IRB # 11–05249).

## Stimuli and experimental design

All subjects performed a semantic judgment task on visually presented stimuli (*Figure 1a*). Stimuli consisted of 70 colored drawings: 36 belonging to the semantic category of living items (e.g., animals, plants) and 34 belonging to the semantic category of nonliving items (e.g., tools, furniture).

To validate the set of stimuli, a behavioral study was conducted on a separate group of 54 age-matched healthy subjects (31 women; 47 right-handed; age = 74.21 years ± 8.63; education = 15 years ± 2.02). First, subjects had to report the most common name for each drawing (i.e., *Identify the item in the image: what is the first name that comes to mind?*). They were given the possibility of providing a second term if needed (i.e., *If appropriate, write the second name that came to mind.*). They were then asked to rate how familiar they are with the item on a 7-point scale from *not at all familiar* to *very familiar*. Finally, they were asked whether the item belongs to the category of living or nonliving items, and to rate how prototypical for that category the item is (i.e., *How good is this picture as example of an item of that category?*) on a 7-point scale from *bad example* to *good example*. Data were collected with Qualtrics software (Qualtrics, Provo, UT. https://www.qualtrics.com) and subjects recruited from the broad pool of subjects enrolled in the above described UCSF MAC

healthy aging cohort. For each stimulus, we calculated the percentage of agreement with our pre-set categorization, average familiarity, average prototypicality, and then compared the living and nonliving categories. For living items, the average percentage of agreement with the assigned category was 96.86% ± 4.07, the lowest score was 75.93% for the item *dinosaur*. For nonliving items, the average percentage of agreement was 99.18% ± 1.20, the lowest score was 96.30% for the items *pizza* and *hamburger*. A two-tailed t-test revealed that the difference between the two categories was significant (p=0.002): the rate of agreement was higher for nonliving items than for living ones. The average prototypicality of living items was 6.24 ± 0.52 (range 6.74–4), while for nonliving items 6.47 ± 0.32 (range 6.85–5.19) for nonliving items. Again, a two-tailed t-test revealed a significant difference between the two categories (p=0.032): nonliving items were judged more prototypical of their category than living ones. As for familiarity, the average for living items was 6.15 ± 0.32 (range 6.8–4.81), while for nonliving items was 6.67 ± 0.21 (range 6.91–6.02). Even in this case the difference between the two categories was significant (two-tailed t-test, p<0.001): nonliving items were judged more familiar.

Images of the two categories were also compared in terms of visual complexity (calculated as Shannon entropy via the python package Scikit-Image, https://scikit-image.org/). No significant difference between living (3.04 ± 0.84) and nonliving (3.13 ± 0.96) items emerged. Finally, we compared stimuli in terms of the length (number of letter), imaginability, concreteness, and familiarity of their most common lexical label as extracted from the Medical Research Council (MRC, http://web-sites.psychology.uwa.edu.au/school/MRCDatabase/uwa_mrc.htm) Psycholinguistic Database, and word frequency was extracted from the Corpus of Contemporary American English (COCA, https://www.wordfrequency.info/). Consistent with our online questionnaire, the only statistically significant differences between the two categories were imageability (living: 613.19 ± 19.62, nonliving: 596.43.15 ± 28.08, p=0.03) and familiarity (living: 498.26 ± 69.32, nonliving: 547.96 ± 45.82, p<0.001). All the psycholinguistic variables characterizing the stimuli are shown in *Table 3*.

Visual stimuli were projected into the magnetically shielded MEG scanner room via a system of mirrors mounted within the scanner room for this purpose, with the final mirror positioned roughly 61 cm from the subject's face. Subjects were instructed to classify the pictures as living or nonliving by pressing one of two response buttons with their dominant hand. Stimuli were displayed for 2 s, with an inter-stimulus interval jittered between 1.7 and 2.1 s. A total of 170 trials were presented:

**Table 3.** Psycholinguistic characteristics of the stimuli.

Stimuli consisted of 70 colored drawings illustrating living items (n = 36) or nonliving items (n = 34). Length, imaginability, concreteness, and familiarity (norm) were extracted from the Medical Research Council (MRC) Psycholinguistic Database searching for the most common label for each item. Similarly, frequency was extracted from the Corpus of Contemporary American English (COCA). Category agreement, category prototypicality, and familiarity (quest.) were assessed with a behavioral study on separate age-matched healthy controls. As a proxy for visual complexity, we used Shannon entropy as computed with Scikit-Image. Values shown are mean (standard deviation). * Indicate values significantly different between the two categories (two-tailed t-test, p<0.05).

|  | Living items | Nonliving items |  |
| --- | --- | --- | --- |
| N | 36 | 34 |  |
| Examples | *Fish, flower* | *Scissors, train* |  |
| Frequency (log) | 3.69 (0.54) | 3.96 (0.65) |  |
| Length (# of letters) | 5.29 (1.58) | 5.61 (1.84) |  |
| Imageability | 613.19 (19.62) | 596.43 (28.08) | * |
| Familiarity (norm) | 498.26 (69.32) | 547.96 (45.82) | * |
| Familiarity (quest.) | 6.15 (0.32) | 6.67 (0.21) | * |
| Concreteness | 608.27 (16.26) | 599.10 (25.94) |  |
| Category agreement | 96.86 (4.07) | 99.18 (1.20) | * |
| Category prototypicality | 6.24 (0.52) | 6.47 (0.32) | * |
| Visual complexity | 3.04 (0.84) | 3.13 (0.96) |  |

each individual stimulus was repeated 2.5 times in a random order. E-Prime (https://pstnet.com/products/e-prime/) was used to present the stimuli; events from E-Prime and the response pad were automatically routed into the imaging acquisition software and integrated with MEG traces in real time.

## Behavioral analyses

Subject performance, that is, reaction times and accuracy, was analyzed using an analysis of variance based on the two stimuli categories (living vs. nonliving) and two cohorts (controls vs. svPPA patients) using the Python statistical library (statsmodels – http://www.statsmodels.org). Data from one outlier in the svPPA cohort were excluded from the behavioral analyses (average reaction times were 1.35 s vs. 0.8 s in the whole cohort).

## MRI protocol and analyses

Structural T1-weighted images were acquired on a 3 T Siemens system (Siemens, Erlagen, Germany) installed at the UCSF Neuroscience Imaging Center, equipped with a standard quadrature head coil with sequences previously described (Mandelli et al., 2014). MRI scans were acquired within 1 year of the MEG data acquisition.

To identify regions of atrophy, svPPA patients were compared to a separate set of 25 HC collected using the same protocol (14 females, mean age 66.2 ± 8.5) via voxel-based morphometry (VBM). Image processing and statistical analyses were performed using the VBM8 Toolbox implemented in Statistical Parametric Mapping (SPM8, Wellcome Trust Center for Neuroimaging, London, UK, http://www.fil.ion.ucl.ac.uk/spm) running under Matlab R2013a (MathWorks). The images were segmented into gray matter, white matter, and CSF, bias corrected, and then registered to the Montreal Neurological Institute (MNI). Gray matter value in each voxel was multiplied by the Jacobian determinant derived from the spatial normalization to preserve the total amount of gray matter from the original images. Finally, to ensure the data are normally distributed and compensate for inexact spatial normalization, the modulated gray matter images were smoothed with a full-width at half-maximum Gaussian kernel filter of $8 \times 8 \times 8$ mm$^3$. A general linear model was then fit at each voxel, with one variable of interest (group) and three confounds of no interest: gender, age, education, and total intracranial volume (calculated by summing across the gray matter, white matter, and CSF images). The resulting statistical parametric map was thresholded at $p<0.05$, with family-wise error correction, and a cluster extent threshold of 100 voxels.

## MEG protocol and analyses

Neuromagnetic recordings were conducted using a whole-head 275 axial gradiometer MEG system (Omega 2000, CTF, Coquitlam, BC, Canada) at a sampling rate of 1200 Hz, under a bandpass filter of 0.001–300 Hz, while subjects performed the task. Subjects were lying supine, with their head supported near the center of the sensor array. Head position was recorded before and after each scan using three fiducial coils (nasion, left/right preauricular) placed on the subject. All subjects included in the current study exhibited movement under 1 cm, as measured pre- and post- experimental run. The two cohorts did not differ in average motion level: svPPA 0.28 cm (SD 0.11), HC 0.39 cm (SD 0.22) (p=0.12). Twenty-nine reference sensors were used to correct distant magnetic field disturbance by calculating a synthetic third-order gradiometer (Warrington and Shallice, 1984; Vrba and Robinson, 2001), which was applied to signal post-acquisition. Datasets were epoched with respect to stimulus presentation onset (stimulus-locked trials from −0.5 to 1.0 s) and artifacts rejected using a semi-automated process outlined as follows: noisy channels were identified as having more than 20 trials exceeding 1.5 pT amplitude under a temporary bandpass filter of 3–50 Hz, with no more than five channels in the sensor array removed. Epochs were then flagged and removed for any remaining artifacts exceeding the 1.5 pT threshold. Mean number of trials included in analyses for the two groups did not significantly differ (svPPA mean = 155 trials [SD = 20, range 121–170], control mean = 162 [SD = 16, range 103–172], two-tailed t[34]=1.059, p=0.297).

Alignment of structural and functional images was performed using three prominent anatomical points (nasion and preauricular points), marked in the individuals' MR images and localized in the MEG sensor array using the three fiducial coils attached to these points during the MEG scan. A 3D grid of voxels with 5 mm spatial resolution covering the entire brain was created for each subject

and recording, based on a multisphere head model of the coregistered structural 3D T1-weighted MR scan. Reconstruction of whole brain oscillatory activity within these voxels was performed via the Neurodynamic Utility Toolbox for MEG (NUTMEG; http://nutmeg.berkeley.edu, *Hinkley et al., 2020*), which implements a time-frequency optimized adaptive spatial filtering technique to estimate the spatiotemporal estimate of neural sources. The tomographic volume of source locations was computed using a 5 mm lead field that weights each cortical location relative to the signal of the MEG sensors (*Dalal et al., 2008*; *Dalal et al., 2011*). The beamforming algorithm choses was a variant of the synthetic aperture magnetometry (SAM) inverse solution (*Vrba and Robinson, 2001*) as implemented in NUTMEG (*Hinkley et al., 2020*). SAM is considered a scalar beamformer as it optimizes dipole orientation at the source to maximize signal.

We sought to focus on induced changes in brain activity, that is, to study modulations of ongoing oscillatory processes that are not necessarily phased-locked (*Makeig et al., 2004*). Moreover, we wished to explore both high- and low-frequency ranges as they bear different functional interpretations, in particular their association with different spatial scales: high- and low-frequency oscillations are associated with local and distributed computations, respectively (*Donner and Siegel, 2011*). Thus, we examined task-related modulations of ongoing oscillatory processes in five frequency bands: theta (3–7 Hz), alpha (8–12 Hz), beta (12–30 Hz), low-gamma (30–55 Hz), and high-gamma (63–117 Hz) (FIR filter having widths of 300 ms for theta/alpha, 200 ms for beta, 150 ms for low-gamma, and 100 ms for high-gamma; sliding over 25 ms time windows). Source power for each voxel location in a specific time window and frequency band was derived through a noise-corrected pseudo-F statistic expressed in logarithmic units (decibels, dB), describing signal magnitude during an 'active' experimental time window relative to an equivalently sized, static pre-stimulus baseline 'control' window (*Robinson and Vrba, 1999*). Single subject beamformer reconstructions were spatially normalized by applying each subject's T1-weighted transformation matrix to their statistical map.

Group analyses were performed, with no additional smoothing, on normalized reconstructions using statistical nonparametric mapping (SnPM; *Singh et al., 2003*), both within-group and between-groups. Three-dimensional average and variance maps across subjects were calculated at each time point and smoothed with a $20 \times 20 \times 20$ mm$^3$ Gaussian kernel (*Dalal et al., 2008*; *Dalal et al., 2011*). From this map, pseudo-t statistics evaluated the magnitude of the contrast obtained at each voxel and time. Voxel labels were permuted to create a t-distribution map for within- and between-group contrasts ($2^N$ permutations, where N = number of subjects, up to 10,000 permutations). Each voxel's t-value was evaluated using $2^N$ degrees of freedom to determine the corresponding p-value associated with each voxel's pseudo-F value (*Singh et al., 2003*). These cortical significance maps were spatially thresholded to include only voxels designated as 'gray matter' within the automated anatomical labeling atlas (*Tzourio-Mazoyer et al., 2002*), and the additional requirement for voxels with uncorrected p-values attaining a threshold of p<0.005 to include 26 adjacent gray matter voxels at p<0.005, effecting a cluster-based threshold of activity. We utilized these maps to examine the pattern of activation during semantic categorization separately for controls and svPPA patients (SnPM one-sample, two-tailed t-test against baseline) and directly compare svPPA patients and controls to highlight spatiotemporal clusters of differential activity between the two cohorts (SnPM two-sample, two-tailed t-test). It should be noted that while we are conducting nonparametric stats with a conservative cluster thresholding to reduce spurious findings from voxel to voxel (p-value threshold 0.005), no correction (nor additional test) is performed to account for multiple time windows or frequency bands.

Finally, we conducted two ROI post hoc analyses. The first one aimed at visualizing, across all frequency bands, the time course of the differential activation between the two cohorts. To avoid circularity and cherry-picking, ROI selection was based on anatomical references justified not only by the whole brain results, but also by the theoretical framework adopted, that is, the hub-and-spoke model (*Ralph et al., 2017*) and the idea of a perceptual-to-conceptual gradient of information processing along the ventral visual path (*Borghesani and Piazza, 2017*). Following previous investigations of the oscillatory dynamics of (visual) semantic processing (*Mollo et al., 2017*; *Clarke et al., 2018*), we draw three spheres of 20 mm radius along the vOT: left occipital pole (OCC, MNI: −10, −94, −16), left vOT (MNI: −50, −52, −20), and left ATL (MNI: −30, −6, −40). *Figure 4* illustrates, for each frequency band, the evolution of single subjects' values across the whole epoch. The second one aimed at characterizing the full time-frequency spectrum in two voxels centered in the OCC ROI

(MNI: −10, −94, −16) and in the peak of activation observed at a cohort level (MNI: −34.8, –93.9, 2.7). For each subject, to extract activity at these specific locations, a broadband covariance matrix was first computed with all trial epoch data. This sample covariance matrix and the column-normalized lead field matrix specific to each voxel was used to calculate a linearly constrained minimum variance spatial filter (*Van Veen et al., 1997*). Broadband source activity for that voxel in each epoch was estimated by applying the spatial filter on the sensor data and projecting along the orientation with the maximum power. The estimated voxel time series was then subject to time-frequency analysis implemented in Fieldtrip (*Oostenveld et al., 2011*), using multi-taper spectral estimation methods. Event-related spectral power changes (2–120 Hz in 1 Hz steps) were estimated from the time-frequency decomposition, by scaling the length of the time window and the amount of frequency smoothing according to the frequency by a factor of 5 and 0.4, respectively (e.g., the time window at 10 Hz is 500 ms, the frequency smoothing 4 Hz). Once all subjects' power spectra had been computed, group averages for both svPPA and HC were calculated and plotted (*Figure 4*).

## Data availability and visualization

The sensitive nature of patients' data and our current ethics protocol do not permit open data sharing. However, anonymized, pre-processed, group-level data used to generate the figures have been uploaded to NeuroVault [https://neurovault.org/collections/FTKQLDFP/]. The clinical and neuroimaging data used in the current paper are available from the senior author (SN), upon formal request indicating name and affiliation of the researcher as well as a brief description of the use that will be done of the data. All requests will undergo UCSF-regulated procedure thus require submission of a Material Transfer Agreement (MTA) which can be found at https://icd.ucsf.edu/material-transfer-and-data-agreements No commercial use would be approved. All images are rendered with the BrainNet Viewer (http://www.nitrc.org/projects/bnv/; *Xia et al., 2013*).

## Acknowledgements

The authors thank the patients and their families for the time and effort they dedicated to this research. Funding: This work was funded by the following National Institutes of Health grants (R01NS050915, K24DC015544, R01NS100440, R01DC013979, R01DC176960, R01DC017091, R01EB022717, R01AG062196). Additional funds include the Larry Hillblom Foundation, the Global Brain Health Institute, and UCOP grant MRP-17–454755. These supporting sources were not involved in the study design, collection, analysis, or interpretation of data, nor were they involved in writing the paper or the decision to submit this report for publication.

## Additional information

### Funding

| Funder | Grant reference number | Author |
| --- | --- | --- |
| National Institutes of Health | R01NS050915 | ML Gorno-Tempini |
| National Institutes of Health | K24DC015544 | ML Gorno-Tempini |
| National Institutes of Health | R01NS100440 | JF Houde |
| National Institutes of Health | R01DC013979 | SS Nagarajan |
| National Institutes of Health | R01DC176960 | SS Nagarajan |
| National Institutes of Health | R01DC017091 | SS Nagarajan |
| National Institutes of Health | R01EB022717 | SS Nagarajan |
| National Institutes of Health | R01AG062196 | SS Nagarajan |
| Larry L. Hillblom Foundation | | ML Gorno-Tempini |
| Global Brain Health Institute | | ML Gorno-Tempini |
| University of California | MRP-17-454755 | SS Nagarajan |

The funders had no role in study design, data collection and interpretation, or the decision to submit the work for publication.

## Author contributions
V Borghesani, Conceptualization, Data curation, Formal analysis, Investigation, Visualization, Methodology, Writing - original draft, Writing - review and editing; CL Dale, Data curation, Formal analysis, Methodology, Writing - review and editing; S Lukic, Conceptualization, Investigation, Writing - review and editing; LBN Hinkley, Resources, Data curation, Software, Formal analysis, Methodology, Writing - review and editing; M Lauricella, W Shwe, D Mizuiri, S Honma, Resources, Methodology; Z Miller, Resources; B Miller, Conceptualization, Funding acquisition, Project administration; JF Houde, Conceptualization, Funding acquisition, Project administration, Writing - review and editing; ML Gorno-Tempini, SS Nagarajan, Conceptualization, Supervision, Funding acquisition, Writing - original draft, Project administration, Writing - review and editing

## Author ORCIDs
V Borghesani (iD) https://orcid.org/0000-0002-7909-8631

## Ethics
Human subjects: The study was approved by the UCSF Committee on Human Research and all subjects provided written informed consent (IRB # 11-05249).

## Decision letter and Author response
Decision letter https://doi.org/10.7554/eLife.63905.sa1
Author response https://doi.org/10.7554/eLife.63905.sa2

# Additional files
## Supplementary files
• Transparent reporting form

## Data availability
The sensitive nature of patients' data and our current ethics protocol do not permit open data sharing. However, anonymized, pre-processed, group-level data used to generate the figures have been uploaded to NeuroVault [https://neurovault.org/collections/FTKQLDFP/]. The clinical and neuroimaging data used in the current paper are available from the Senior Author (S.N.), upon formal request indicating name and affiliation of the researcher as well as a brief description of the use that will be done of the data. All requests will undergo UCSF regulated procedure thus require submission of a Material Transfer Agreement (MTA) which can be found at https://icd.ucsf.edu/material-transfer-and-data-agreements No commercial use would be approved.

The following datasets were generated:

| Author(s) | Year | Dataset title | Dataset URL | Database and Identifier |
|---|---|---|---|---|
| Borghesani V | 2021 | Neural dynamics of semantic categorization in semantic variant of Primary Progressive Aphasia | https://neurovault.org/collections/FTKQLDFP/ | NeuroVault, FTKQLDFP/ |
| Borghesani V | 2021 | Incoming MTAs/Outgoing Human MTAs/Data Agreements | https://icd.ucsf.edu/material-transfer-and-data-agreements | Material Transfer Agreement, agreements |

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
