## [Decision Letter]

**Acceptance summary:**

This study investigates how dysfunction in the anterior temporal lobe (ATL) alters dynamic activity during semantic categorization. Magnetoencephalography (MEG) responses were contrasted between patients with semantic variant Primary Progressive Aphasia (svPPA) and age-matched healthy controls. Despite similar profiles of behavioural performance on the categorization task, the svPPA patients showed enhanced γ synchronization in the occipital lobe compared to controls suggesting an increased engagement of early perceptual mechanisms for completing the task, as opposed to semantic identification of the picture.

**Decision letter after peer review:**

Thank you for submitting your article "Neural dynamics of semantic categorization in semantic variant of Primary Progressive Aphasia" for consideration by *eLife*. Your article has been reviewed by 3 peer reviewers, and the evaluation has been overseen by Chris Baker as Reviewing and Senior Editor. The following individuals involved in review of your submission have agreed to reveal their identity: Alex Clarke (Reviewer #2); Aneta Kielar (Reviewer #3).

The reviewers have discussed the reviews with one another and the Reviewing Editor has drafted this decision to help you prepare a revised submission.

Summary:

Borghesani and colleagues aimed to understand how dysfunction in the anterior temporal lobe (ATL) alters dynamic activity during semantic categorization. They contrast MEG responses between 18 patients with semantic variant Primary Progressive Aphasia (PPA) and 18 age-matched healthy controls. Both groups show similar profiles of behavioural performance on the task, and broad similarities in MEG responses. Critically, however, svPPA patients show enhanced γ synchronization in the occipital lobe compared to controls. The authors interpret this as reflecting increased engagement of / reliance on early perceptual mechanisms for completing the task, as opposed to semantic identification of the picture.

Overall, the reviewers found the manuscript interesting. As svPPA is a rare (but scientifically informative) disorder, the sample size is impressive, and given that relatively few MEG studies exist in PPA at all, this is an interesting dataset. However, the general opinion is that the results could be more fully characterized, which would allow for more expansive interpretations and inferences.

Essential revisions:

1) Statistical thresholding

Using a high threshold prevents false positives, but may also lead to false negatives, and that may be the case here, with the high threshold contributing to an unrealistic impression of spatial specificity in MEG. It is obvious from the average responses in both groups that these oscillatory responses are widespread through the brain. Indeed, the α and β responses are significant in the majority of cortical voxels. This basic property of the responses should be presented clearly and prominently in the paper – not just in supplementary information where only a minority of readers will even see it. The authors then use an extremely high and conservative statistical threshold to contrast differences between the two groups. P<.005 uncorrected is a highly conservative threshold already, even before cluster-thresholding is added (although with data as smooth as MEG beamforming solutions, cluster-thresholding is unlikely to change anything). Essentially, this makes only the strongest part of the activation survive, and while it is valid to conclude that a significant group difference exists (protected from Type 1 error), this can also give a false impression that the difference is specific to that region. A more realistic characterization of the results would involve measuring differences in the strength of the responses between groups on a broader level, possibly the sensors or in large ROIs – and not ROIs pre-selected to show a dramatic difference by first searching the whole brain for the most significant effects – that is the classic "double-dipping" fallacy in neuroimaging.

2) Frequency bands

The ERD/ERS in each frequency band is treated as a separate entity, ignoring the fact that these bands are arbitrary and frequency is a continuous quantity. This matters because much is made of the fact that svPPA participants exhibited greater ERS in the low-γ range, and that this was correlated with reaction time. Supplementary Figure 1 shows that both groups had strong occipital ERS in the high-γ range, but only svPPA showed it in the low γ range as well. This suggests that the ERS in the svPPA group may simply have been shifted to a lower frequency range. A more fulsome characterization of these group differences via time-frequency analysis and/or power spectral analysis would help clarify what is going on here.

3) Decreased responses in svPPA?

It is surprising that svPPA participants only exhibited increased MEG responses compared to controls – assuming that both γ ERS and β ERD can be interpreted as increased neural activation, which is a reasonable assumption based on the literature. No decreases in the svPPA group are found, and thus the observed increases can be plausibly attributed to compensatory processes as framed by the authors. However, certain analysis choices may play a role in producing this data pattern. In particular, the authors state (line 611): "To remove potential artifacts due to neurodegeneration or eye movement (lacking electrooculograms), we masked statistical maps using patients' ATL atrophy maps (see section MRI protocol and analyses), as well as a ventromedial frontal mask."

It is not clear whether this masking was conducted in group space from average atrophy maps, or on an individual level. In either case, this is not well justified. What is the physical mechanism by which tissue undergoing neurodegeneration can be said to generate an artifactual signal? Atrophied tissue still contains living neurons with ionic currents; these are real signals not artifacts, and furthermore, atrophy is a continuous process with tissue further from the epicenter also undergoing similar neurodegenerative mechanisms. Atrophied tissue may well generate electromagnetic signals that are different from healthy tissue, and such differences should be included in this paper. There may be regions of hypoactivation as well as hyperactivation in this svPPA group. If the hypoactivation localizes to atrophied tissue and the hyperactivation to other regions, that will bolster the case that we are seeing compensatory processes, but it isn't certain with half the story masked. The statistical masking of the frontal region is also not really a valid solution to eye movement artifacts. The authors would have to present evidence that the region that they masked corresponds to the region potentially affected by eye movements. However, many studies have found that beamforming already does a pretty good job of removing ocular artifact from estimated brain signals, except for very close to the eyes.

4) RT correlation

The correlation with reaction time in the occipital cortex is consistent with the idea that the ERS there may reflect compensatory overreliance on perceptual information, but it isn't conclusive. The authors suggest that svPPA patients are able to categorize the stimuli correctly based on visual features, but are unable to name them. What about testing for correlations with the out-of-scanner behavioural measures that established that the patients have a naming deficit? It would strengthen the case if atrophy or hypoactivation (see comment above) correlated with the naming deficit.

5) Neural dynamics

As the paper is about 'Neural dynamics', this aspect could be developed, with the timing of the effects characterized further, and considered more in relation to the conclusions. For example, the main finding is the increased occipital γ response in svPPA compared to controls. Looking at Figure 3, there is a peak in the svPPA group near 200 ms, and very little synchronized activity in the control group. This is interesting as there are many ways we could have seen svPPA > controls, but this suggests that the γ synchronization response associated with compensation is specific to the svPPA group (and largely absent from controls – also from Supp Figure 1), and is distinguished from an initial visual evoked response (peaking ~100 ms). We recommend discussing and characterizing the dynamics of this effect more, such as what a later occipital effect could tell us about dynamics given ATL dysfunction? Is this increase a result of a lack of top-down effects from ATL?

6) Low-level vs. High-level

The occipital γ effect looks like the primary visual cortex, which might suggest the effects are not related to higher-level perceptual features (such as has eyes, teeth) as the authors suggest, but rather low-level visual effects. Do the authors perhaps think the effects could relate to enhanced processing of visual details (as related to the ideas of Hochstein and Asher's reverse hierarchy), or whether the effects relate to additional visual input following a visual saccade?

7) VBM

The VBM results for the svPPA patients were surprising given that all the atrophy appeared in the left hemisphere. There can be hemispheric differences in svPPA, but is this a true lateral pattern (meaning the right ATL is intact) or a product of VBM being run so that the most atrophied hemisphere is shifted to the left side? If the VBM maps are correct, and the svPPA patients are only showing left hemisphere atrophy, then what does this suggest about the role of the right ATL, and the bilateral nature of occipital increased in svPPA?

8) Task performance

Both svPPA patients and healthy controls achieved around 80% accuracy in the categorization task. This seems surprisingly low given, (1) the task (living vs. nonliving after seeing the image for 2 seconds), (2) that all the images were pretested and had high name agreement, and (3) that items were repeated on average 2.5 times. Is there something that explains this low performance for all individuals?

9) Compensation

One question for clarification is whether the recruitment of the occipital areas in svPPA is truly "compensatory", does it indicate a shift of resources due to the anterior temporal atrophy. Is the recruitment of the parieto-occipital regions associated with more accurate performance?

10) Other frequency bands (related to point 2 above)

The main results concentrate on the differences between patient and controls in the low γ range. There are also significant effects in the other frequency bands (e.g., high γ, β and α). What is the functional significance of these effects?

[Editors' note: further revisions were suggested prior to acceptance, as described below.]

Thank you for submitting your article "Neural dynamics of semantic categorization in semantic variant of Primary Progressive Aphasia" for consideration by *eLife*. Your article has been reviewed by 3 peer reviewers, and the evaluation has been overseen by Chris Baker as the Reviewing and Senior Editor. The following individuals involved in review of your submission have agreed to reveal their identity: Jed A Meltzer (Reviewer #1); Alex Clarke (Reviewer #2).

The reviewers have discussed their reviews with one another, and the Reviewing Editor has drafted this to help you prepare a revised submission. In general, the reviewers are still positive about the manuscript but think that the claims need to be tempered slightly and would like to see the time-frequency dynamics presented in more detail (as requested in the original reviews).

Essential Revisions:

1) Further analysis of the time-frequency dynamics is needed as laid out in the reviewers' comments below.

2) While the findings are consistent with a compensatory interpretation, especially given the equivalent performance in both groups, other interpretations are also possible. This should be discussed more fully, and the discussion could be grounded in earlier literature that has considered similar compensatory accounts e.g. age differences – for example many papers by Cheryl Grady show that older adults have more bilateral activation than younger. Those results were considered in the context of what kinds of findings constitute evidence of compensation vs. pathology.

Reviewer #1:

The revision by Borghesani et al., is much improved in terms of technical procedures and description, and most of the concerns raised by the reviewers have been adequately addressed. It is an interesting finding in a somewhat rare patient group.

I really only have one remaining concern that I still think should be addressed.

This paper puts a lot of emphasis on a particular interpretation of changes in oscillatory dynamics between the svPPA group and the control group. Based mainly on one particular finding – increased low-γ ERS in the occipital cortex for the svPPA group, the authors argue that svPPA patients compensate for their conceptual impairments by increasing their reliance on early perceptual processing implemented in occipital areas. Originally this interpretation was supported by both the increased low-γ ERS and also a correlation with performance. Since the changed analysis procedures resulted in dropping the claim of correlation, everything now rests on the shoulders of that low-γ finding. I think it needs to be unpacked a bit more.

If the increased low-γ finding were unambiguously interpretable as "activation" or "recruitment," this would be a straightforward story. But MEG data is complex and nuanced, more so than fMRI in my opinion, and there are some nuances here that are being overlooked. Both groups have robust activation in a higher band, high-γ, a band which is more strongly linked to increased neural firing and increased BOLD than the low-γ band is. On the other hand, the patients appear to have somewhat less ERD in the β band in this area, and β ERD is also strongly linked to neural firing and BOLD. The low γ band is kind of tricky – sometimes it goes up, sometimes it goes down. To understand this more, it would definitely help to see a real time-frequency decomposition of the activity, at least in this one key area.

We asked for this in the first round of review, and the authors declined to do it, citing concerns about time-frequency resolution tradeoff. That is not very convincing – there is ample resolution available in this data to characterize the effect in both time and frequency, and anyway in this case it is really frequency that raises the important questions – the group difference lasts for at least 400 ms so fine temporal resolution isn't so necessary. The authors argue that a lack of significant difference for the high γ band argues against a "frequency shift" interpretation – perhaps "spread" would be a more precise term than shift; in any case, it is clear that frequency is a key dimension in the difference of oscillatory response between these two groups, and it needs to be characterized better given the importance of this finding.

Perhaps a more practical concern is that the authors used optimized beamforming weights for specific frequency bands, precluding a traditional broad-band time-frequency analysis. However, they can still characterize time-frequency reactivity using an additional post-hoc analysis. This could be done on the sensor level, which I understand the authors do not prefer for legitimate reasons, but it could also be done in source space with non-frequency-optimized beamforming weights. This may not afford the same spatial resolution, but the blob of differential γ activity between groups is very large; precise spatial resolution isn't needed to answer this question.

I also think that given this ambiguity in the central finding, the authors should soften their conclusions somewhat and offer alternative interpretations. There is certainly a difference in the occipital lobe between groups, and that is interesting, but the idea that it's a compensatory increase in the patient group is somewhat speculative – consistent with the data, but not proven.

Reviewer #2:

I've read through all the comments and review responses, and think overall the manuscript is improved and several points made clearer.

I think there are a few points that remain for me:

1. The source analysis procedure is clear, along with thresholding and cluster extent. Yet, I didn't see any information on how the authors control for the effects over the sliding time windows, or for the frequency bands? We're these statistical contrasts taken into account?

2. New ROI data is presented showing the effects in 3 regions and across the frequency bands, with the authors claiming a difference in low γ activity around 100 ms. Yet stating the effect is around 100 ms doesn't seem to capture the data in the plot. It looks like difference may first appear around 100 ms, but peak nearer 200 ms, and continue throughout the epoch. I think a fuller description is warranted.

3. The ATL is no longer masked out from any of the analysis, and I would state this somewhere for clarity. There is also apparent signal coming from the atrophy region – mainly in β and α – it might be worth commenting on this.

4. Finally, to avoid switching back between Figures 2/3 and Table 3, I would consider adding if the effects relate to ERS or ERD in the table.

Reviewer #3:

I thank authors for addressing reviewers' comments. I think that the manuscript has improved. I don't have further comments.

---

## [Author Response]

Essential revisions:1) Statistical thresholdingUsing a high threshold prevents false positives, but may also lead to false negatives, and that may be the case here, with the high threshold contributing to an unrealistic impression of spatial specificity in MEG. It is obvious from the average responses in both groups that these oscillatory responses are widespread through the brain. Indeed, the α and β responses are significant in the majority of cortical voxels. This basic property of the responses should be presented clearly and prominently in the paper – not just in supplementary information where only a minority of readers will even see it.

We thank the Reviewers for highlighting the value of the within-group whole brain activations. The information previously provided by our supplementary figure is now included as the first main figure in the manuscript (see updated Figure 2).

The authors then use an extremely high and conservative statistical threshold to contrast differences between the two groups. P<.005 uncorrected is a highly conservative threshold already, even before cluster-thresholding is added (although with data as smooth as MEG beamforming solutions, cluster-thresholding is unlikely to change anything). Essentially, this makes only the strongest part of the activation survive, and while it is valid to conclude that a significant group difference exists (protected from Type 1 error), this can also give a false impression that the difference is specific to that region.

While we agree that our approach has been rather conservative, we hold that this is warranted to avoid detection of false positives and focus on meaningful group-level effects. We would also like to note that the anonymized, pre-processed, group-level data used to generate the figures have been uploaded to NeuroVault [https://neurovault.org/collections/FTKQLDFP/]. This allows direct exploration of the results to further assess the regional specificity of the effects we here report and discuss.

A more realistic characterization of the results would involve measuring differences in the strength of the responses between groups on a broader level, possibly the sensors or in large ROIs – and not ROIs pre-selected to show a dramatic difference by first searching the whole brain for the most significant effects – that is the classic "double-dipping" fallacy in neuroimaging.

First, we would like to stress that our analyses do look for differences at the broadest level (whole-brain, all frequencies and time windows). The post-hoc region of interest analysis was conducted with the sole purpose of investigating the relation between subjects’ behavioral performance and neural activity in the two clusters that showed the strongest effects.

However, we agree with the Reviewers that our approach was inherently arbitrary and that generally speaking ROI-selection can lead to circularity and cherry-picking (not to mention multiple comparisons issues if one was to test out many different ROIs). We recognize that a better approach is that of showing group-level time plots across all frequencies in a-priori defined ROIs. Our final selection of three ventral occipito-temporal ROIs is consistent with the results of our whole brain analyses while being theory-driven. It builds on the theoretical framework we adopted (the hub-and-spoke model, Lambon-Ralph et al., 2016) and the previously put forward hypothesis of a perceptual-to-conceptual gradient of information processing along the ventral visual path (Borghesani and Piazza, 2017). Following, Mollo et al., 2011 and Clarke et al., 2018, we built spheres of 20 mm radius in three locations previously associated with visual and semantic processing: occipital pole (OCC, MNI: −10, −94, −16), left ventral occipitotemporal cortex (VOT, MNI:−50, −52, −20), and left ATL (MNI: −30, −6, −40), see updated Figure 4.

Finally, we would argue that a sensor level analysis approach will not yield (additional) meaningful insights. Any MEG sensor has broad sensitivity to signals arising from many brain regions. Therefore, it is very difficult to evaluate sensor level information in terms of brain topographies. In contrast, our analyses in brain source space following source reconstruction allows for clear interpretation of the findings in terms of brain topography.

2) Frequency bandsThe ERD/ERS in each frequency band is treated as a separate entity, ignoring the fact that these bands are arbitrary and frequency is a continuous quantity. This matters because much is made of the fact that svPPA participants exhibited greater ERS in the low-γ range, and that this was correlated with reaction time. Supplementary Figure 1 shows that both groups had strong occipital ERS in the high-γ range, but only svPPA showed it in the low γ range as well. This suggests that the ERS in the svPPA group may simply have been shifted to a lower frequency range. A more fulsome characterization of these group differences via time-frequency analysis and/or power spectral analysis would help clarify what is going on here.

We agree with the Reviewers that frequency bands are inherently arbitrary. The ability to resolve time-frequency components in ERD/ERS are governed by time-frequency trade-off principles. To obtain, higher frequency resolution requires longer-time windows for analyses which in-turn will limit the temporal resolution of the source reconstructions. Conversely, to obtain high-temporal resolution one will forsake frequency resolution. The approach we have taken in relation to the time-frequency trade-off is three-fold: (1) Time-frequency optimized source reconstruction, where the source reconstruction adaptive filters are optimized for a particular time and frequency resolution, (2) Pre-specified frequency bands based on the literature, and (3) Variable time-windows for different frequency bands (100 ms for high-γ and 300 ms for α bands). A consequence of this approach indeed is that we are not able to examine the frequency content with higher resolution. Alternative approaches in the literature have been to forsake temporal resolution for higher frequency resolution, as suggested by the Reviewers. However, an unintended consequence of such an approach will result in non-time-frequency optimized source reconstruction with reduced sensitivity to higher frequency bands like γ band.

Critically, our results cannot be reduced to a shift in frequency range as specific spatio-temporal clusters are detected across the various bands. For instance, focusing on the occipital low γ effect mentioned by the Reviewers, arguably the strongest result, one should note that the svPPA > HC in low γ is not associated with a HC > svPPA in high γ. Rather than the interpretation suggested by the Reviewers, this observation suggests that computations in svPPA (but not HC) are occupying both low and high γ ranges.

Finally, as discussed above, our post-hoc ROI analyses now include all frequency bands, allowing further comparison of the full spectrum of time-frequency effects (see updated Figure 4).

3) Decreased responses in svPPA?It is surprising that svPPA participants only exhibited increased MEG responses compared to controls – assuming that both γ ERS and β ERD can be interpreted as increased neural activation, which is a reasonable assumption based on the literature. No decreases in the svPPA group are found, and thus the observed increases can be plausibly attributed to compensatory processes as framed by the authors.

First of all, we would like to stress that all our analyses are two-tailed, allowing for effects in either direction to emerge. We have now clarified this aspect in the method section.

Second, we would like to point out that not all significant clusters denote increased activity in svPPA (vs. HC). We apologize for the poor visualization choices of the original Figure 2 (see now updated Figure 3), but few (smaller) clusters of HC>svPPA are detected, for instance, in the β band (late in the epoch, in left middle temporal gyrus and superior frontal gyrus, see negative t-values in Table 3).

Third, we would like to highlight that smaller, weaker, effects for HC > svPPA should not come as a surprise. As discussed in our manuscript, higher activations for svPPA (vs. controls) are the most common finding in the few previous functional neuroimaging studies comparing these populations, see for instance PET results from Mummery et al., 1999, fMRI results from Wilson et al., 2009, MEG results from Pineault et al., 2019.

However, certain analysis choices may play a role in producing this data pattern. In particular, the authors state (line 611): "To remove potential artifacts due to neurodegeneration or eye movement (lacking electrooculograms), we masked statistical maps using patients' ATL atrophy maps (see section MRI protocol and analyses), as well as a ventromedial frontal mask."It is not clear whether this masking was conducted in group space from average atrophy maps, or on an individual level. In either case, this is not well justified. What is the physical mechanism by which tissue undergoing neurodegeneration can be said to generate an artifactual signal? Atrophied tissue still contains living neurons with ionic currents; these are real signals not artifacts, and furthermore, atrophy is a continuous process with tissue further from the epicenter also undergoing similar neurodegenerative mechanisms. Atrophied tissue may well generate electromagnetic signals that are different from healthy tissue, and such differences should be included in this paper. There may be regions of hypoactivation as well as hyperactivation in this svPPA group. If the hypoactivation localizes to atrophied tissue and the hyperactivation to other regions, that will bolster the case that we are seeing compensatory processes, but it isn't certain with half the story masked.

We agree with the Reviewers that the field currently lacks proper strategies to deal with atrophy while attempting to source localize effects in neurodegenerative patients. Please note that the first and main issue is that of defining atrophy itself: atrophic tissue will vary subject by subject (and it is not a all-or-none phenomenon) yet a threshold would need to be established.

In previous work, we took the parsimonious approach of masking out atrophic regions from group level statistics to avoid uninterpretable results (e.g. Borghesani et al., 2020). In other settings, one option is that of correcting region-of-interest statistics including GM volume as covariate (e.g. Ranasinghe et al., 2017). Finally, some would simply report whether electrophysiological differences and atrophy maps overlap or not (e.g., Kielar et al., 2018). Clearly, the underlying issue of how tissue undergoing neurodegeneration affects source modeling is an open problem that requires further exploration.

The statistical masking of the frontal region is also not really a valid solution to eye movement artifacts. The authors would have to present evidence that the region that they masked corresponds to the region potentially affected by eye movements. However, many studies have found that beamforming already does a pretty good job of removing ocular artifact from estimated brain signals, except for very close to the eyes.

We agree with the Reviewers that beamforming already minimizes the effect of ocular artifact (see Bardouille, Picton and Ross, 2006. Correlates of eye blinking as determined by synthetic aperture magnetometry. Clin. Neurophysiol. 117, 952–958.), and that in absence of proper electrooculograms there are no ideal procedures to verify whether ventro-medial frontal signals are spurious or not.

Thus, following Reviewers suggestion of minimizing data manipulation to avoid false negative results (while keeping a conservative approach minimizing false positive ones), we have repeated all our analyses removing the atrophy masking and frontal area masking steps. The current results are obtained simply ensuring that the statistical maps include only voxels designated as grey matter within the AAL atlas (Tzourio-Mazoyer et al., 2002). Overall, the current approach leads to the same results with the added advantage of being more parsimonious.

As our current visualization allows to appreciate (see updated Figure 3), the effects we observe are largely outside atrophied areas. Clearly, the issue of the electrophysiological effects of neurodegeneration is open and warrants further dedicated investigations.

4) RT correlationThe correlation with reaction time in the occipital cortex is consistent with the idea that the ERS there may reflect compensatory overreliance on perceptual information, but it isn't conclusive. The authors suggest that svPPA patients are able to categorize the stimuli correctly based on visual features, but are unable to name them. What about testing for correlations with the out-of-scanner behavioural measures that established that the patients have a naming deficit? It would strengthen the case if atrophy or hypoactivation (see comment above) correlated with the naming deficit.

We thank the Reviewers for this suggestion. However, to avoid circularity/cherry-picking in the ROI selection, as advised by the Reviewers, we have now drastically modified our approach. The current post-hoc analyses are conducted on a-priori regions centered around coordinates taken from the literature rather than those obtained from group differences in the current study (see updated figure 4). Within these ROIs, we do not observe any correlation between reaction times and neural signal (in either cohort) and thus have removed the correlation analysis from the Results section.

Unfortunately, further investigation of brain-behavior correlations are severely hampered by (1) the size of our sample (i.e., only 18 data points in each group); (2) the fact that patients were not explicitly tested for naming on the stimuli used in the MEG classification task; (3) the fact that controls are at ceiling in the Boston Naming Task while patients show (as expected) frank impairment but not sufficient variability (see Table 1). Consistent with these issues, we find that BNT performance does not correlate with the performance measures obtained in the semantic classification task (neither RTs nor accuracy).

5) Neural dynamicsAs the paper is about 'Neural dynamics', this aspect could be developed, with the timing of the effects characterized further, and considered more in relation to the conclusions. For example, the main finding is the increased occipital γ response in svPPA compared to controls. Looking at Figure 3, there is a peak in the svPPA group near 200 ms, and very little synchronized activity in the control group. This is interesting as there are many ways we could have seen svPPA > controls, but this suggests that the γ synchronization response associated with compensation is specific to the svPPA group (and largely absent from controls – also from Supp Figure 1), and is distinguished from an initial visual evoked response (peaking ~100 ms). We recommend discussing and characterizing the dynamics of this effect more, such as what a later occipital effect could tell us about dynamics given ATL dysfunction? Is this increase a result of a lack of top-down effects from ATL?

We thank the Reviewers for this comment. We believe that the major changes to the analytical steps now implemented, see major #1, address this issue as well. Namely, a better characterization of the full breath of neural dynamics, across the epoch and the frequencies, is now enabled by (1) the removal of the atrophy mask, (2) having put within-group findings as main result, see Figure 2, (3) the choice of 3 a-priori ROIs, and (4) the presentation of the group results in those regions across all frequencies.

The result section “Post-hoc region-of-interest analyses” now highlights how the main difference between svPPA patients and healthy controls is the heightened low γ activity over the occipital region, evident around 100 ms post stimuli onset but maintained across the whole epoch. These findings rule out an explanation of the observed whole brain differences as mere temporal (or frequency) shift while highlighting the spectral (and spatial) specificity of the main results. For a discussion on how these findings can be interpreted in terms of functional compensation, please see major point #9.

6) Low-level vs. High-levelThe occipital γ effect looks like the primary visual cortex, which might suggest the effects are not related to higher-level perceptual features (such as has eyes, teeth) as the authors suggest, but rather low-level visual effects. Do the authors perhaps think the effects could relate to enhanced processing of visual details (as related to the ideas of Hochstein and Asher's reverse hierarchy), or whether the effects relate to additional visual input following a visual saccade?

We thank the Reviewers for giving us the opportunity of clarifying this aspect of our results. We would like to point out that given the regional specificity of MEG source reconstructions, our findings are not specific to primary visual cortex and most certainly include non-primary visual areas. We did not measure eye movements during the task and trials with very large eye-movement artifacts were removed by our preprocessing procedure. Therefore, we cannot quantify (nor compare) the number of saccades across cohorts. Please see also point # 1 and revised manuscript section “Post-hoc region-of-interest analyses” for a discussion on the regional specificity of our findings.

We appreciate the reference to the Reverse Hierarchy Theory (RHT), which was mainly proposed for visual perceptual learning, framing it as a top-down guided process, which begins at high-level areas of the visual system, and progresses backwards to the input levels as/if needed. If, as a “leap-of-faith, we view the changes in processing of visual semantic information in svPPA during the course of the disease as a maladaptive learning process, we would speculate that RHT would predict results that are consistent with our observation – namely that when higher-level areas of the visual system are impaired, the processing shifts to lower-levels along the visual cortical hierarchy. However, given our paradigm it is impossible to contrast heighten bottom-up information (i.e., additional visual input) vs. top-down enhanced processing. Dedicated empirical paradigms are needed to directly investigate the role of feedback connections in modulating information processing. Furthermore, recent empirical studies investigating top-down vs. bottom-up flow of information along the ventral visual path (Dijkstra et al., 2020) question the static nature of visual processing (in either direction) which is the basis of the RHT theory, Dijkstra’s findings associated perception with cycles of recurrent processing while detecting only one feed-back flow during imagery. Interestingly, the 11 Hz oscillation associated with the iteratively update of precepts along the visual hierarchy is consistent with the idea that γ-band activity is linked to bottom-up information while α and β bands with top-down (Fries, 2015). Generally speaking, we do not deem appropriate to further expand the discussion of our results in a top-down vs. bottom-up framework as our findings cannot really speak to this debate.

7) VBMThe VBM results for the svPPA patients were surprising given that all the atrophy appeared in the left hemisphere. There can be hemispheric differences in svPPA, but is this a true lateral pattern (meaning the right ATL is intact) or a product of VBM being run so that the most atrophied hemisphere is shifted to the left side? If the VBM maps are correct, and the svPPA patients are only showing left hemisphere atrophy, then what does this suggest about the role of the right ATL, and the bilateral nature of occipital increased in svPPA?

We apologize if our choice of threshold and colormap suggested a stronger-than-expected left lateralization of svPPA patients’ atrophy. As it can be appreciated now in Figure 1c [as well as in the unthresholded SPM T map loaded NeuroVault, ID 441824] the atrophy is bilateral yet clearly asymmetrical. As a matter of fact, at least 13 svPPA patients present marked left-lateralized atrophy with only 5 of them showing a bilateral or right-lateralized pattern. This percentage of L vs R and the overall pattern is in line with multiple previous evidence from our group as well as others (Binney et al., 2016; Snowden et al., 2018; Woollams and Patterson, 2018).

A comparison of L vs R predominant ATL damage, investigating if (and how) that would affect the observed bilateral occipital hyperactivation would be of interest. Unfortunately, in our sample we don’t have enough R-predominant cases to perform a sound comparison. Given the current understanding of the graded, bilateral, nature of the ATL semantic hub, with the R hemisphere appearing specialized for nonverbal, socio-emotional material, it could be speculated that predominantly R svPPA would have a worst performance, possibly linked to higher (attempt to compensate via) occipital activation. However, we believe that such (perhaps heightened) occipital activation would still have a bilateral presentation: to our knowledge, there are no reasons to believe that low-level visual analysis of the stimuli would show any lateralization.

8) Task performanceBoth svPPA patients and healthy controls achieved around 80% accuracy in the categorization task. This seems surprisingly low given, (1) the task (living vs. nonliving after seeing the image for 2 seconds), (2) that all the images were pretested and had high name agreement, and (3) that items were repeated on average 2.5 times. Is there something that explains this low performance for all individuals?

We are very grateful to the Reviewers for pointing this inconsistency out. While total percentage accuracy was correctly computed out of the 170 trials, an error in our code was incorrectly computing the percentage accuracy condition-wise (living vs. nonliving). The error has been fixed and both the text and figure updated with the correct values. Please note that the mistake did not change the main observation of no statistical difference between cohorts (nor condition, nor their interaction): HC: living: 97.1±6.6, nonliving: 96.8±6.6; svPPA: living: 91.5±6.2, nonliving: 95.9±8.1.

9) CompensationOne question for clarification is whether the recruitment of the occipital areas in svPPA is truly "compensatory", does it indicate a shift of resources due to the anterior temporal atrophy. Is the recruitment of the parieto-occipital regions associated with more accurate performance?

We agree with the Reviewers that an interpretation of our findings as functional compensation is hampered by the disclosed and discussed limitations of our paradigm and sample. Namely, (1) the size of our sample; (2) the similar, almost at ceiling, performance during the classification task of patients and HC; and (3) the fact that patients were not explicitly tested for naming on the same stimuli. Please note that, with the novel ROIs, we do not observe any correlation between reaction times and neural signal, thus cannot provide any explicit evidence of higher occipital activation being associated with better performance.

We would still hold that our patients’ semantic deficit is well established (Table 1) and its origin in ATL neurodegeneration is clear (Figure 1c): the link between svPPA patients neuropsychological and neuroanatomical profiles is undisputed. It is also clear from our data that svPPA patients can perform the semantic classification task as well as healthy controls (Figure 1b), thus somehow overcoming their semantic loss. This behavioral effect is matched, at the neural level, with the spatiotemporal clusters of differential activity we describe (and discuss).

10) Other frequency bands (related to point 2 above)

The main results concentrate on the differences between patient and controls in the low γ range. There are also significant effects in the other frequency bands (e.g., high γ, β and α). What is the functional significance of these effects?

We thank the Reviewers for highlighting the breath of our results: indeed, the occipital low γ findings are the strongest, but other spatiotemporal clusters of interest emerge across all frequencies.

For further discussions of our approach and the resulting frequency-specific results, please see also major point #2 (i.e., ruling out simple time or frequency shift) and #6 (i.e., speculations on top-down vs. bottom-up effects), as well as minor point #4 (i.e., speculation on the local vs. global integration effects).

[Editors' note: further revisions were suggested prior to acceptance, as described below.]

The reviewers have discussed their reviews with one another, and the Reviewing Editor has drafted this to help you prepare a revised submission. In general, the reviewers are still positive about the manuscript but think that the claims need to be tempered slightly and would like to see the time-frequency dynamics presented in more detail (as requested in the original reviews).

We thank the Editor and the Reviewers for their feedback on our revised manuscript. We appreciate the time they invested and their thoughtful comments. We have now included a thorough time-frequency analysis to rule out potential confounds from frequency shift effects that may manifest as power changes. We have now also better framed our speculative interpretation of the results in the discussion. We thus believe that the attached revised paper responds to the few remaining concerns and, thanks to their suggestions, is greatly improved.

Essential Revisions:1) Further analysis of the time-frequency dynamics is needed as laid out in the reviewers' comments below.

We acknowledge that the characterization of the full time-frequency dynamics over the area of interest would allow better appreciation of the observed effect. We now provide in Figure 5 the full time-frequency spectrum in two voxels: one located at the centroid of the OCC ROI (MNI: −10, −94, −16) and a second one located at the peak of activation observed at a cohort level, (MNI: -34.8 -93.9 2.7). For each subject, to extract activity at these specific locations, a broad-band covariance matrix was first computed with all trial epoch data. This sample covariance matrix and the column-normalized lead field matrix specific to each voxel was used to calculate a linearly constrained minimum variance spatial filter (LCMV, Van Veen et al., 1997). Broadband source activity for that voxel in each epoch was estimated by applying the spatial filter on the sensor data and projecting along the orientation with the maximum power. The estimated voxel time-series was then subject to time-frequency analysis implemented in fieldtrip (www.fieldtriptoolbox.org) using multi-taper spectral estimation methods. Event-related spectral power changes (2 to 120 Hz in 1-Hz steps) were estimated from the time–frequency decomposition, by scaling the length of the time window and the amount of frequency smoothing according to the frequency by a factor of 5 and 0.4 respectively (so for instance the time window at 10 Hz is 500ms, the frequency smoothing 4 Hz). Group averages for both svPPA and HC were calculated (using ft_freqgrandaverage) and plotted (using ft_singleplotTFR). The results, shown in Figure 5 , clearly rule out the possibility that observed effects (reported in Figure 2 and 3) can arise from frequency shift or spread. Instead, we observe broad and sustained increase in low-γ band power in svPPA patients that is not seen in healthy controls.

2) While the findings are consistent with a compensatory interpretation, especially given the equivalent performance in both groups, other interpretations are also possible. This should be discussed more fully, and the discussion could be grounded in earlier literature that has considered similar compensatory accounts e.g. age differences – for example many papers by Cheryl Grady show that older adults have more bilateral activation than younger. Those results were considered in the context of what kinds of findings constitute evidence of compensation vs. pathology.

We thank the Reviewers and the Editor for pointing out previous work on age-related differences and for suggesting a more thorough framing of our results within the broader “maintenance, reserve, and compensation” spectrum. We have now revised our discussion to include the following observation.

“Finally, our interpretation of higher low-γ as a sign of compensation is speculative and lies on two-fold inference: that more low-γ band in svPPA means more activity, and that more activity means compensation. Previous literature in healthy and pathological aging suggests that higher activation can be associated with compensatory effects or reflect neuropathology (e.g., Elman et al., 2014). Critically, when considering progressive phenomena such as neurodegeneration one has to acknowledge that hyper- and hypo- activations might reflect different stages of disease and that their relation to behavioral performance might follow a nonlinear U-shaped trajectory (e.g., Gregory et al., 2017). Following Cabeza and colleagues (2018), we believe that greater activity can be interpreted as compensation if two criteria are met: (1) there has to be evidence of a “supply-demand gap”, and (2) there has to be evidence of a beneficial effect on cognitive performance. Our data clearly fulfills the first criterion as svPPA patients present insufficient neural resources (i.e., relatively focal ATL atrophy) and suffer from its behavioral consequences (i.e., pervasive semantic loss). Our findings also fulfill the second criterion as svPPA patients are able to perform the task with both accuracy and reaction times comparable with the healthy controls. Thus, we believe that the neural and behavioral evidence we provide is enough to rule out alternatives to compensation such as inefficiency or pathology. While further studies are warranted to shed light onto the relation between neurodegeneration, neurophysiological markers, and behavior, we hold that the most appropriate (albeit speculative) interpretation of our findings is in terms of compensation.”

Reviewer #1:The revision by Borghesani et al., is much improved in terms of technical procedures and description, and most of the concerns raised by the reviewers have been adequately addressed. It is an interesting finding in a somewhat rare patient group.I really only have one remaining concern that I still think should be addressed.This paper puts a lot of emphasis on a particular interpretation of changes in oscillatory dynamics between the svPPA group and the control group. Based mainly on one particular finding – increased low-γ ERS in the occipital cortex for the svPPA group, the authors argue that svPPA patients compensate for their conceptual impairments by increasing their reliance on early perceptual processing implemented in occipital areas. Originally this interpretation was supported by both the increased low-γ ERS and also a correlation with performance. Since the changed analysis procedures resulted in dropping the claim of correlation, everything now rests on the shoulders of that low-γ finding. I think it needs to be unpacked a bit more.If the increased low-γ finding were unambiguously interpretable as "activation" or "recruitment," this would be a straightforward story. But MEG data is complex and nuanced, more so than fMRI in my opinion, and there are some nuances here that are being overlooked. Both groups have robust activation in a higher band, high-γ, a band which is more strongly linked to increased neural firing and increased BOLD than the low-γ band is. On the other hand, the patients appear to have somewhat less ERD in the β band in this area, and β ERD is also strongly linked to neural firing and BOLD. The low γ band is kind of tricky – sometimes it goes up, sometimes it goes down. To understand this more, it would definitely help to see a real time-frequency decomposition of the activity, at least in this one key area.We asked for this in the first round of review, and the authors declined to do it, citing concerns about time-frequency resolution tradeoff. That is not very convincing – there is ample resolution available in this data to characterize the effect in both time and frequency, and anyway in this case it is really frequency that raises the important questions – the group difference lasts for at least 400 ms so fine temporal resolution isn't so necessary. The authors argue that a lack of significant difference for the high γ band argues against a "frequency shift" interpretation – perhaps "spread" would be a more precise term than shift; in any case, it is clear that frequency is a key dimension in the difference of oscillatory response between these two groups, and it needs to be characterized better given the importance of this finding.Perhaps a more practical concern is that the authors used optimized beamforming weights for specific frequency bands, precluding a traditional broad-band time-frequency analysis. However, they can still characterize time-frequency reactivity using an additional post-hoc analysis. This could be done on the sensor level, which I understand the authors do not prefer for legitimate reasons, but it could also be done in source space with non-frequency-optimized beamforming weights. This may not afford the same spatial resolution, but the blob of differential γ activity between groups is very large; precise spatial resolution isn't needed to answer this question.

Please see the additional analysis and figure 5.

I also think that given this ambiguity in the central finding, the authors should soften their conclusions somewhat and offer alternative interpretations. There is certainly a difference in the occipital lobe between groups, and that is interesting, but the idea that it's a compensatory increase in the patient group is somewhat speculative – consistent with the data, but not proven.

This has been explicitly addressed in the limitations section.

Reviewer #2:I've read through all the comments and review responses, and think overall the manuscript is improved and several points made clearer.I think there are a few points that remain for me:1. The source analysis procedure is clear, along with thresholding and cluster extent. Yet, I didn't see any information on how the authors control for the effects over the sliding time windows, or for the frequency bands? We're these statistical contrasts taken into account?

We’d like to clarify that we are conducting non-parametric stats with cluster thresholding to reduce spurious findings from voxel to voxel, adopting a more conservative p-value threshold (.005) than is typically reported, but with no correction of p values (nor additional tests) to account for multiple time windows or frequency bands. This information has been made explicit in the method section.

2. New ROI data is presented showing the effects in 3 regions and across the frequency bands, with the authors claiming a difference in low γ activity around 100 ms. Yet stating the effect is around 100 ms doesn't seem to capture the data in the plot. It looks like difference may first appear around 100 ms, but peak nearer 200 ms, and continue throughout the epoch. I think a fuller description is warranted.

This has been explicitly added to the Results section.

3. The ATL is no longer masked out from any of the analysis, and I would state this somewhere for clarity. There is also apparent signal coming from the atrophy region – mainly in β and α – it might be worth commenting on this.

This has been explicitly addressed in the limitations section.

4. Finally, to avoid switching back between Figures 2/3 and Table 3, I would consider adding if the effects relate to ERS or ERD in the table.

The suggested change has been implemented.